# PROBING CLIP'S COMPREHENSION OF 360-DEGREE TEXTUAL AND VISUAL SEMANTICS

## ABSTRACT

The dream of instantly creating rich 360-degree panoramic worlds from text is rapidly becoming a reality, yet a crucial gap exists in our ability to reliably evaluate their semantic alignment. Contrastive Language-Image Pre-training (CLIP) models, standard AI evaluators, predominantly trained on perspective image-text pairs, face an open question regarding their understanding of the unique characteristics of 360-degree panoramic image-text pairs. This paper addresses this gap by first introducing two concepts: *360-degree textual semantics*, semantic information conveyed by explicit format identifiers, and *360-degree visual semantics*, invariant semantics under horizontal circular shifts. To probe CLIP's comprehension of these semantics, we then propose novel evaluation methodologies using keyword manipulation and horizontal circular shifts of varying magnitudes. Rigorous statistical analyses across popular CLIP configurations reveal that: (1) CLIP models effectively leverage explicit textual identifiers, demonstrating an understanding of 360-degree textual semantics; and (2) CLIP models fail to robustly preserve semantic alignment under horizontal circular shifts, indicating limited comprehension of 360-degree visual semantics. To address this limitation, we propose a LoRA-based fine-tuning framework that explicitly instills invariance to circular shifts. Our fine-tuned models exhibit improved comprehension of 360-degree visual semantics, though with a slight degradation in original semantic evaluation performance, highlighting a fundamental trade-off in adapting CLIP to 360-degree panoramic images.

## 1 INTRODUCTION

360-degree panoramic images, typically represented by the equirectangular projection (Ai et al., 2025; da Silveira et al., 2022; Yan et al., 2024), provide comprehensive $360° \times 180°$ views of scenes, making them essential in various applications such as virtual reality (Brivio et al., 2021), gaming (Fan et al., 2019), and immersive media (Weissig et al., 2012). Traditionally, capturing high-quality 360-degree panoramas requires specialized equipment and professional expertise, which are prohibitively expensive and technically challenging for non-expert users. This limitation motivates the exploration of alternative methods to simplify and democratize the creation of panoramic content.

Recent advancements in text-to-image (T2I) generative models (Nichol et al., 2022; Podell et al., 2023; Ramesh et al., 2022; Rombach et al., 2022), which are trained on large-scale paired image-text datasets such as LAION-400M and LAION-5B (Schuhmann et al., 2021; 2022), have enabled researchers to adapt pre-trained T2I models (Rombach et al., 2022) to synthesize diverse and photorealistic 360-degree panoramas directly from natural language descriptions (Feng et al., 2023; Kalischek et al., 2025; Wang et al., 2024a; 2023; Zhang et al., 2024). These methods significantly lower the entry barriers to producing panoramic content and facilitate novel applications (Ma et al., 2024; Wang & Xue, 2025; Yang et al., 2024; Zhou et al., 2024).

A pivotal aspect of advancing these text-driven generative systems is the accurate evaluation of semantic alignment between generated 360-degree panoramic images and their corresponding textual prompts. Contrastive Language-Image Pre-training (CLIP) models (Ilharco et al., 2021; Radford et al., 2021) have become the *de facto* standard for evaluating image-text semantic alignment (Hessel et al., 2021). These models embed images and text prompts into a shared semantic space, where cosine similarity between their respective embeddings quantifies alignment. However, the predominant

"*a 360 degree view of* a bathroom with a toilet, sink and shower"    [horizontally circular- shifted version]     "*a modern indoor swimming pool with lights*"    [horizontally circular-shifted version]

(a) 360-Degree Panoramic Image-Text Pair      (b) Perspective Image-Text Pair

Figure 1: Example of two types of image-text pairs. Textually, the explicit format identifier is highlighted (left in each pair). Visually, the corresponding horizontally circular-shifted versions are shown (right in each pair).

training of CLIP models on perspective image-text pairs (Radford et al., 2021) raises questions about their applicability to evaluating 360-degree panoramic image-text pairs, which present fundamentally different characteristics (see Fig. 1).

360-degree panoramic image-text pairs exhibit distinct semantic attributes in both textual and visual modalities. Textually, prompts for such images often include explicit 360-degree panoramic format identifiers (e.g., "*a 360 degree view of*", "*360 photo*"), which convey what we define as **360-degree textual semantics**. Visually, images capture a complete spherical ($360° \times 180°$) view, which results in inherent semantic invariance under horizontal circular shifts; the scene content remains identical despite rotation. We term this invariant semantics **360-degree visual semantics**. These unique visual and textual attributes motivate our central research question: **To what extent can standard CLIP models, predominantly trained on perspective image-text pairs, comprehend the distinct semantics inherent in 360-degree panoramic image-text pairs?**

To answer this, we conduct a systematic investigation into CLIP's understanding of these two semantic types using curated datasets of real and synthesized 360-degree panoramic images. Our analysis, grounded in rigorous statistical hypothesis testing, probes the capabilities of popular CLIP configurations (ViT-B/32, ViT-B/16, and ViT-L/14).

First, to assess 360-degree textual semantics, we establish a keyword manipulation approach. Specifically, we measure how the presence or absence of explicit panoramic identifiers in textual prompts affects CLIP's image-text alignment. By comparing CLIP scores between original prompts containing identifiers and modified prompts where identifiers are replaced with generic cues (e.g., "*photo*", "*image*"), we test whether models leverage format-specific textual cues. Results show that across all configurations, CLIP scores are significantly higher when explicit identifiers are present. These findings provide strong evidence that CLIP models effectively comprehend and exploit 360-degree textual semantics, underscoring the critical role of such identifiers in textual prompts for achieving accurate semantic evaluation of 360-degree panoramic images with CLIP models.

Second, to probe 360-degree visual semantics, we evaluate whether CLIP maintains stable alignment between textual prompts and 360-degree panoramic images under horizontal circular shifts of varying magnitudes. For each image, we generate shifted versions and compute CLIP scores with the original text prompt. A robust understanding of 360-degree visual semantics would require these scores to remain stable across shifts. To formalize this, we define a **stability bound**, derived from CLIP's response to a canonical semantic-preserving transformation, namely the horizontal flip (Wang et al., 2024b), and computed by using Tukey's boxplot method (Tukey et al., 1977). We then test whether the absolute differences between the original and shifted scores exceed this bound. Our results reveal that CLIP lacks a robust grasp of 360-degree visual semantics.

To address this limitation, we propose a fine-tuning framework designed to explicitly instill invariance to horizontal circular shifts in CLIP models. Our approach employs Low-Rank Adaptation (LoRA) (Hu et al., 2022) applied to the image encoder, guided by a specialized loss function with a balancing parameter that jointly enforces invariance to shifts while regularizing the CLIP model to preserve its original semantic predictions. Experimental results demonstrated that our fine-tuned models acquired a robust understanding of 360-degree visual semantics. However, this enhancement introduces a slight degradation in the original semantic evaluation capability of CLIP, revealing a fundamental trade-off between enhanced comprehension of 360-degree visual semantics and preservation of baseline semantic performance.

The contributions of this work can be summarized as follows: (1) We introduce and define two novel concepts: **360-degree textual semantics** (semantic information conveyed by explicit format identifiers) and **360-degree visual semantics** (invariant semantics under horizontal circular shifts). We further design targeted evaluation methodologies to probe CLIP's comprehension of these semantics.

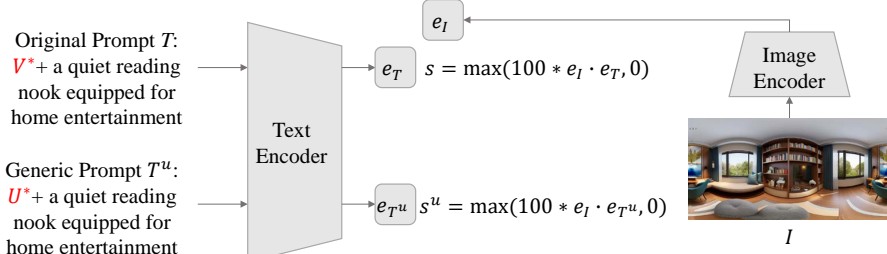

Figure 2: Overview of our framework to evaluate CLIP models' understanding of 360-degree textual semantics. The format cue $V^*$ is a keyword explicitly identifying the 360-degree panoramic image format (e.g., "*360 panorama*", "*360 photo*"), while $U^*$ is a generic cue (e.g., "*photo*", "*image*") that lacks specific 360-degree panoramic format information.

(2) Rigorous statistical analysis shows that all the evaluated CLIP models benefit significantly from explicit textual format identifiers, confirming their effective use of 360-degree textual semantics. This emphasizes the importance of including format-specific cues in prompts for accurate semantic evaluation of 360-degree panoramas. (3) We provide the first systematic statistical evidence that CLIP models fail to robustly preserve semantic alignment under horizontal circular shifts, highlighting their limited understanding of 360-degree visual semantics. (4) We propose a LoRA-based fine-tuning framework that successfully enhances CLIP models' comprehension of 360-degree visual semantics, while also revealing the trade-off between this enhanced capability and the models' original performance.

## 2 ANALYSIS OF CLIP MODELS

360-degree panoramic image-text pairs possess unique semantics within both the textual and visual modalities. Textually, the prompts contain explicit format identifiers (e.g., "*a 360 degree view of*", "*360 photo*"). We conceptualize the semantic information conveyed by these specific textual cues, indicating the 360-degree panoramic nature of the image, as **360-degree textual semantics**. Visually, the images inherently capture a complete $360° \times 180°$ spherical view of an entire scene. A key consequence of this spherical geometry is their semantic invariance under horizontal circular shifts; the scene content remains identical, merely rotated horizontally. We denote this invariant property as **360-degree visual semantics**. To investigate whether CLIP models effectively comprehend these distinct semantics, we propose evaluation methodologies targeting each aspect.

### 2.1 PROBING UNDERSTANDING OF 360-DEGREE TEXTUAL SEMANTICS

To evaluate CLIP's ability to capture 360-degree textual semantics, we propose a method based on keyword manipulation. This approach assesses the model's capability to comprehend format identifiers within text prompts associated with 360-degree panoramic images.

Let $I$ denote a 360-degree panoramic image and $T$ its corresponding textual description. As illustrated in Fig. 2, $T$ can be decomposed into a **format cue** and a **content descriptor**. The format cue, denoted as $V^*$, is a concise keyword or phrase explicitly identifying the 360-degree panoramic image format (e.g., "*360 panorama*", "*360 photo*" or "*a 360 degree view of*"). The content descriptor conveys the semantic and visual elements of the scene. We construct a generic prompt $T^u$ by replacing only $V^*$ in the original prompt $T$ with a generic cue $U^*$ (e.g., "*photo*","*image*") that lacks specific 360-degree panoramic format information.

Using a pre-trained CLIP model, we extract the normalized image embedding $e_I$ from $I$, and normalized text embeddings $e_T$ and $e_{T^u}$ from the original prompt $T$ and the generic prompt $T^u$, respectively. We then compute the CLIP scores:

$$s = \max(100 * e_I \cdot e_T, 0), \; s^u = \max(100 * e_I \cdot e_{T^u}, 0), \tag{1}$$

where $s$ quantifies the alignment between $I$ and the original panoramic description, while $s^u$ reflects the alignment with the generic description.

**Statistical Hypothesis Test** If a CLIP model effectively leverages 360-degree-specific textual cues, the presence of the explicit format identifier $V^*$ in $T$ should strengthen the image-text association

Table 1: **[OpenCLIP, LAION-400M]** [$V^* =$"*<360panorama>,* "], the Wilcoxon Signed-Rank test (Wilcoxon, 1945) results for different CLIP models on the two paired image-text datasets (*360_real* and *360_syn*) defined in Sec. 4.1, where the null hypothesis is the original score $s$ is not greater than the generic score $s^u$, and the significance level ($\alpha$) is 0.01. The p-values less than $\alpha$ are in bold.

| ViT | $U^* =$"" | | | | $U^* =$"*image,* " | | | |
| | *360_real* | | *360_syn* | | *360_real* | | *360_syn* | |
| | *statistic* | *p-value* | *statistic* | *p-value* | *statistic* | *p-value* | *statistic* | *p-value* |
| B/32 | 2847676 | **0** | 2847270 | **0** | 2847655 | **0** | 2845186 | **0** |
| B/16 | 2847596 | **0** | 2663038 | **0** | 2847057 | **0** | 2219021 | **0** |
| L/14 | 2847691 | **0** | 2847516 | **0** | 2847667 | **0** | 2847196 | **0** |

compared with the generic prompt $T^u$, resulting in $s > s^u$. To test this, we conduct a one-sided superiority test on the paired differences, $\{s_i - s_i^u\}_{i=1}^M$, collected from all $M$ 360-degree panoramic image-text pairs in our dataset. The null hypothesis ($H_0$) posits that the format-specific cue offers no semantic benefit, meaning that the paired difference is not greater than zero:

$$H_0 : s - s^u \leq 0. \tag{2}$$

A rejection of this null hypothesis would provide strong evidence that the model effectively comprehends and exploits the 360-degree textual semantics.

**Findings**  Table 1 shows the results of statistical tests on three CLIP models using two different generic cues $U^*$ ("" and "*image,* ") to replace the format cue $V^*$ ("*<360panorama>,* "). All p-values are consistently below the significance level (0.01). This leads to a decisive rejection of the null hypothesis, providing compelling evidence that these evaluated CLIP models effectively discern and utilize 360-degree textual semantics.

## 2.2 PROBING UNDERSTANDING OF 360-DEGREE VISUAL SEMANTICS

To investigate CLIP models' comprehension of 360-degree visual semantics (i.e., the invariant semantics under horizontal circular shifts), we propose an evaluation methodology centered on horizontal circular shifts of varying magnitudes.

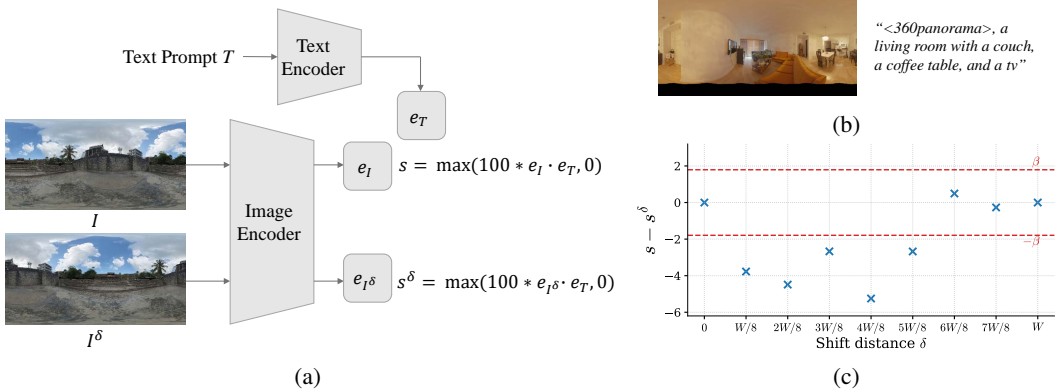

Figure 3: (a) Overview of our framework to assess CLIP models' understanding of 360-degree visual semantics. Image $I^\delta$ is obtained by applying a horizontal circular shift of $\delta$ pixels to $I$ of size $H \times W$; (b) Example of a 360-degree panoramic image-text pair and (c) its CLIP score differences ($s - s^\delta$) using ViT-B/32 across diverse shift distances, where stability bound $\beta = 1.7919$.

Let $I$ denote a 360-degree panoramic image of size $H \times W$, paired with its textual description $T$ (see Fig. 3a). We generate a shifted version, $I^\delta$, by applying a horizontal circular shift of $\delta$ pixels to $I$, where $\delta \in \{1, 2, \ldots, W - 1\}$. Due to the spherical nature, $I^\delta$ depicts the identical scene from the same viewpoint as $I$, differing only by a rotation around the vertical axis. Given a CLIP model, we extract the normalized text embedding $e_T$ from $T$ via the text encoder, and the normalized image embeddings $e_I$ and $e_{I^\delta}$ from the original image $I$ and the shifted image $I^\delta$, respectively, via the image encoder. Subsequently, we compute the two CLIP scores:

$$s = \max(100 * e_I \cdot e_T, 0), \quad s^\delta = \max(100 * e_{I^\delta} \cdot e_T, 0), \tag{3}$$

where $s$ and $s^\delta$ quantify the semantic alignment for the original and shifted images, respectively.

To systematically probe the model's robust comprehension of this invariant semantics, we apply a set of $N-1$ distinct horizontal circular shifts with magnitudes $\delta_j$ given by $\delta_j = \frac{j \cdot W}{N}$ for $j \in \{1, 2, \dots, N-1\}$. For each of these shifted 360-degree panoramic images, we compute its CLIP score with the original text prompt ($T$), yielding a set of scores: $\{s^{W/N}, s^{2W/N}, \dots, s^{(N-1)W/N}\}$.

**Statistical Hypothesis Test**   A CLIP model possessing a robust understanding of 360-degree visual semantics should produce highly stable CLIP scores across the full spectrum of horizontal circular shifts. This implies that the magnitude of the difference between $s$ and $s^\delta$ should be negligibly small. To formally test for this stability, we conduct a series of one-sided hypothesis tests. For a specific shift magnitude $\delta_j$, we first calculate the absolute score differences, $\{|s_i - s_i^{\delta_j}|\}_{i=1}^{M}$, from all $M$ 360-degree panoramic image-text pairs in our dataset. We then define a **stability bound** $\beta > 0$, representing the maximum score change considered practically insignificant. The null hypothesis ($H_{0,j}$) for this specific shift is that the model is not stable, meaning that the absolute difference is greater than or equal to the stability bound:

$$H_{0,j} : |s - s^{\delta_j}| \geq \beta. \tag{4}$$

This test is performed independently for each of the $N-1$ shift magnitudes. The CLIP model can be deemed to possess a robust understanding of 360-degree visual semantics only if the null hypothesis of non-stability ($H_{0,j}$) is rejected for all shifts.

**Definition of the Stability Bound**   A critical component of our stability test is the definition of a principled, non-arbitrary stability bound $\beta$. To establish a data-driven value, we anchor our bound to the CLIP model's response to a canonical semantics-preserving transformation: the horizontal flip (Wang et al., 2024b). Specifically, we calculate the absolute difference in CLIP scores, $|s_i - s_i^{flip}|$, for each of the $M$ image-text pairs in our dataset, where $s_i^{flip}$ is the score of the horizontally flipped image. To derive a threshold that is both robust and adaptive to the model's inherent score variance, we adopt the standard method for outlier detection used in a Tukey boxplot (Tukey et al., 1977). We define $\beta$ as the upper fence of the distribution of these absolute differences. Specifically, we first compute the first quartile ($Q1$) and the third quartile ($Q3$) of these absolute differences. The interquartile range (IQR) is then IQR = $Q3 - Q1$. Our stability bound is formally defined as

$$\beta = Q3 + 1.5 \times \text{IQR}. \tag{5}$$

This method defines "insignificant change" based on the model's own behavior, providing a fair and model-specific benchmark for stability.

Table 2: **[OpenCLIP, LAION-400M]**, the Wilcoxon Signed-Rank test results under horizontal circular shift of various $\delta_j$ pixels for different CLIP models on the ***360_real*** dataset, where the null hypothesis is that $|s - s^{\delta_j}|$ is greater than or equal to the stability bound $\beta$, and the significance level ($\alpha$) is 0.01. The p-values less than $\alpha$ are in bold.

| ViT | $\beta$ | $\delta_j$ | $W/8$ | $2W/8$ | $3W/8$ | $4W/8$ | $5W/8$ | $6W/8$ | $7W/8$ |
|-----|---------|-----------|-------|--------|--------|--------|--------|--------|--------|
| B/32 | 1.7919 | *statistic* | 967086 | 1558059 | 1745605 | 1781527 | 1723312 | 1513504 | 969987 |
|      |        | *p-value* | **0** | 1 | 1 | 1 | 1 | 0.9961 | **0** |
| B/16 | 1.6547 | *statistic* | 1106536 | 1724595 | 1906849 | 1918511 | 1897948 | 1700325 | 1117369 |
|      |        | *p-value* | **0** | 1 | 1 | 1 | 1 | 1 | **0** |
| L/14 | 1.4245 | *statistic* | 1233361 | 1937346 | 2120371 | 2196213 | 2163110 | 1930355 | 1257236 |
|      |        | *p-value* | **0** | 1 | 1 | 1 | 1 | 1 | **0** |

**Findings**   Table 2 reports the results of our statistical tests on three different CLIP models. We do not find sufficient evidence to reject the null hypothesis across all seven shifts simultaneously. Therefore, we conclude that these evaluated CLIP models do not possess a robust understanding of 360-degree visual semantics. To further illustrate this lack of stability, we present a 360-degree panoramic image-text pair in Fig. 3b, and plot the score differences ($s - s^\delta$) across various shift distances using ViT-B/32 in Fig. 3c, which clearly shows some differences outside the bound.

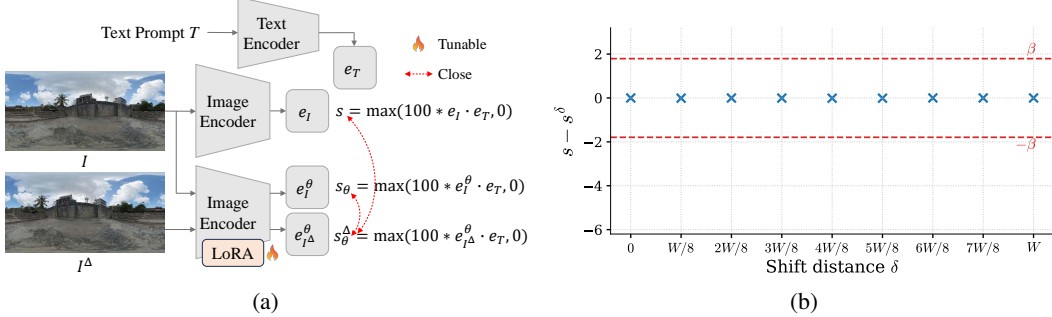

(a)            (b)

Figure 4: (a) Overview of the fine-tuning framework using LoRA. (b) CLIP score differences $(s - s^\delta)$ for the same pair in Fig. 3b using fine-tuned ViT-B/32, where stability bound $\beta = 1.7919$.

## 3   IMPROVING COMPREHENSION OF 360-DEGREE VISUAL SEMANTICS

To address this limitation identified in Sec. 2.2, we design a fine-tuning framework to instill an explicit understanding of 360-degree visual semantics in pre-trained CLIP models. Our approach, illustrated in Fig. 4a, employs Low-Rank Adaptation (LoRA) (Hu et al., 2022) to fine-tune only the image encoder of the CLIP model. The fine-tuning process is guided by a specialized loss function:

$$L_{FT} = \lambda \cdot L_{charb}(s_\theta^\Delta, s_\theta) + (1 - \lambda) \cdot L_{charb}(s_\theta^\Delta, s), \tag{6}$$

where $L_{charb}(x, y) = \sqrt{(x - y)^2 + \epsilon^2}$ denotes the Charbonnier loss (Charbonnier et al., 1994), a smoothed L1 penalty, with $\epsilon$ empirically set to $1 \times 10^{-3}$.

This loss combines two components: (1) an *invariance term* $L_{charb}(s_\theta^\Delta, s_\theta)$, which minimizes the difference between the score of an original panoramic image ($s_\theta$) and its circularly shifted version ($s_\theta^\Delta$) with a randomly selected shift distance $\Delta \in \{0, 1, 2, \ldots, W - 1\}$, both computed by the fine-tuned model; and (2) a *regularization term* $L_{charb}(s_\theta^\Delta, s)$, which minimizes the difference between the score of the shifted image produced by the fine-tuned model ($s_\theta^\Delta$) and that of the original image produced by the frozen pre-trained model ($s$). The weighting parameter $\lambda \in (0, 1)$ balances these two terms, thereby controlling the trade-off between enforcing invariance to horizontal circular shifts and preserving the semantic predictions of the original CLIP model. Fig. 4b shows the score differences of the same image-text pair in Fig. 3b, evaluated with the fine-tuned ViT-B/32. In this case, all score differences fell within the stability bound, visually confirming the enhanced comprehension of 360-degree visual semantics achieved through fine-tuning.

## 4   EXPERIMENTS

### 4.1   DATASETS AND MODELS

**Paired Image-Text Datasets.** To support our evaluations, we constructed two paired image-text datasets: *360_real* and *360_syn*. The process began by generating base textual descriptions for 2,438 real-world 360-degree panoramic images (1024×512 resolution) sourced from Laval Indoor (Gardner et al., 2017) and Laval Outdoor (Hold-Geoffroy et al., 2019), using BLIP-2 (Li et al., 2023). These automatically generated descriptions, however, exhibited two main issues: (1) the presence of directional cues (e.g., "*in the middle*"), which could bias the evaluation; and (2) inconsistent 360-degree panoramic format identifiers (see Fig. 5). To address the first issue, we filtered out images whose associated descriptions contained such directional cues, resulting in a refined subset of 2,386 images. For the second issue, we standardized the textual descriptions of the remaining 2,386 images by employing ChatGPT (OpenAI, 2025) to remove the panoramic format identifiers from the base prompts. Subsequently, each standardized description was prepended with the string "*<360panorama>,* ", which is the default input processing of Diffusion360 (Feng et al., 2023) and which we designate as the format cue $V^*$ in this study. The resulting 2,386 augmented prompts, paired with their corresponding real-world 360-degree panoramic images, formed the *360_real* dataset. These augmented prompts were also input into the Diffusion360 generator, producing 2,386 360-degree panoramic images (1024×512 resolution), which, paired with their augmented prompts, constituted the *360_syn* dataset. A flowchart to produce the two paired datasets is provided in Sec. B.1.

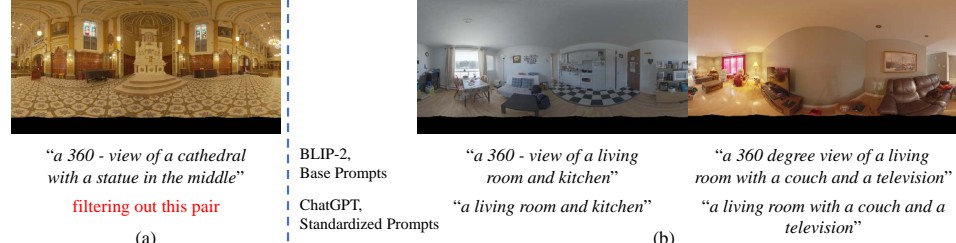

*"a 360 - view of a cathedral with a statue in the middle"*

filtering out this pair

(a)

BLIP-2, Base Prompts

ChatGPT, Standardized Prompts

*"a 360 - view of a living room and kitchen"*

*"a living room and kitchen"*

*"a 360 degree view of a living room with a couch and a television"*

*"a living room with a couch and a television"*

(b)

Figure 5: (a) Example of an image-text pair containing the directional cue ("*in the middle*"). (b) Examples of base prompts and standardized prompts from BLIP-2 and ChatGPT.

**Models of Interest and Implementation Details.** The CLIP (Ilharco et al., 2021; Radford et al., 2021) model comprises two core components: a text encoder and an image encoder. The text encoder is typically based on the Transformer (Vaswani et al., 2017) architecture. For the image encoder, Vision Transformer (ViT) backbones (Dosovitskiy et al., 2020) are commonly adopted, as they exhibit superior performance and efficiency compared to ResNet-based alternatives (He et al., 2016). Standard ViT configurations commonly used in influential CLIP models include ViT-B/32, ViT-B/16 and ViT-L/14, where 'ViT-X/Y' indicates a ViT of size $X$ ('B' for Base, 'L' for Large) with a patch size of $Y \times Y$ pixels. Given the widespread adoption and representative role of the three CLIP variants (Fang et al., 2023; Sun et al., 2023; Xu et al., 2023; Zhai et al., 2023), this study focuses on evaluating their capabilities in comprehending 360-degree visual and textual semantics. For clarity and consistency throughout this paper, these specific CLIP models will be referred to by their respective image encoder configurations: ViT-B/32, ViT-B/16, and ViT-L/14.

The CLIP models evaluated in this main paper are sourced from OpenCLIP (Ilharco et al., 2021), trained on the LAION-400M (Schuhmann et al., 2021) dataset. To further demonstrate the generalization of our findings and proposed fine-tuning framework, we report the results of CLIP models from OpenCLIP trained on the LAION-2B (Schuhmann et al., 2022) dataset in Sec. F and results for the original OpenAI CLIP models (Radford et al., 2021) in Sec. G. In our evaluation, the number of equal divisions ($N$) is set to 8. The width ($W$) of 360-degree panoramic images is 1024, while the total number of image-text pairs ($M$) is 2386. All of the experiments in this paper were performed on an RTX A6000 GPU.

## 4.2 RESULTS FOR 360-DEGREE TEXTUAL SEMANTICS

**CLIP models can comprehend 360-degree textual semantics.** To implement the statistical hypothesis test outlined in Sec. 2.1, we first determined whether a parametric paired t-test (Student, 1908) or a non-parametric Wilcoxon Signed-Rank test (Wilcoxon, 1945) was appropriate. The results of the normality test, conducted using the Shapiro-Wilk test (Shapiro & Wilk, 1965), are presented in Table 4 (Sec. C.1), which motivated the use of the non-parametric Wilcoxon Signed-Rank test for our one-sided superiority test.

We then examined whether CLIP models benefit from 360-degree panoramic textual cues using the *360_real* and *360_syn* datasets. Following the approach introduced in Sec. 2.1, the specific format cue ("$<360panorama>,$ ") in the original prompts was replaced with two different generic cues ("" and "*image,* "), producing the generic prompts $T^u$ along with their corresponding CLIP scores $s^u$. These scores were then compared with the original scores $s$. The results of these analyses, listed in Table 1, indicate that all evaluated CLIP models successfully capture 360-degree textual semantics. Further experiments with additional generic cues ("*photo,* " and "*picture,* ") and format cues ("*a 360 degree view of,* " and "*360 photo,* "), detailed in Appendix C, reinforce this conclusion. These consistent findings demonstrate that all evaluated CLIP models effectively discern and utilize 360-degree panorama-specific textual cues, resulting in significantly stronger image-text alignment when such cues are present. Consequently, these results underscore the considerable importance of incorporating 360-degree panoramic identifiers in textual prompts to enable a more accurate assessment of corresponding 360-degree image content.

## 4.3 RESULTS OF 360-DEGREE VISUAL SEMANTICS

**CLIP models lack a robust understanding of 360-degree visual semantics.** We first established appropriate statistical tests for visual semantics analysis. Normality assessments of the paired absolute

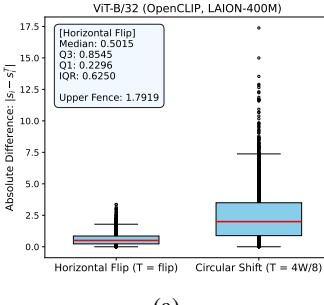 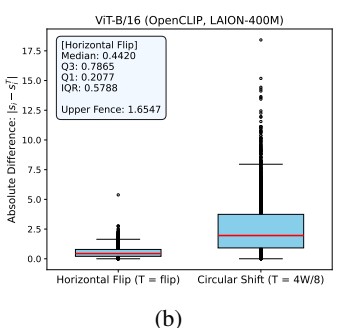 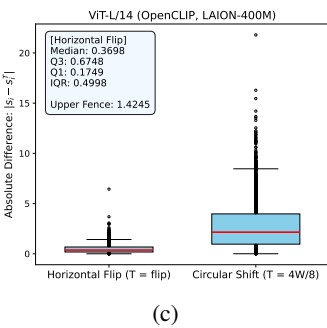

(a)  (b)  (c)

Figure 6: Boxplots of absolute score differences ($|s_i - s_i^T|$) under two diverse transformations for three various CLIP models on the ***360_real*** dataset.

Table 3: [**Fine-Tuned, OpenCLIP, LAION-400M**]. The rest caption is as for Table 2.

| $\lambda$ | ViT | $\beta$ | $\delta_j$ | $W/8$ | $2W/8$ | $3W/8$ | $4W/8$ | $5W/8$ | $6W/8$ | $7W/8$ |
|---|---|---|---|---|---|---|---|---|---|---|
| 0.9889 | B/32 | 1.7919 | *statistic* | 0 | 0 | 0 | 0 | 0 | 0 | 0 |
| | | | *p-value* | **0** | **0** | **0** | **0** | **0** | **0** | **0** |
| 0.9899 | B/16 | 1.6547 | *statistic* | 0 | 0 | 0 | 0 | 0 | 0 | 0 |
| | | | *p-value* | **0** | **0** | **0** | **0** | **0** | **0** | **0** |
| 0.9919 | L/14 | 1.4245 | *statistic* | 0 | 0 | 0 | 0 | 0 | 0 | 0 |
| | | | *p-value* | **0** | **0** | **0** | **0** | **0** | **0** | **0** |

score differences ($|s - s^{\delta_j}|$), as presented in Table 10 (Sec. D), again indicated violations of parametric assumptions, which led us to employ the Wilcoxon Signed-Rank test.

Following the method outlined in Sec. 2.2, we present the distribution of absolute score differences ($|s_i - s_i^{flip}|$) across three CLIP models on the left side of each subfigure in Fig. 6. The stability bounds $\beta$ for ViT-B/32, ViT-B/16, and ViT-L/14 are also reported in the corresponding text boxes. Using these stability bounds, we conducted one-sided Wilcoxon Signed-Rank tests on the ***360_real*** dataset for CLIP models trained on LAION-400M (see Sec. 2.2). The results are reported in Table 2, indicating that none of the evaluated CLIP models exhibits a robust understanding of 360-degree visual semantics. The results on the ***360_syn*** dataset are reported in Table 11 (Sec. D), which further confirms this conclusion.

To provide insight into these findings, Fig. 6 also compares boxplots of absolute score differences under horizontal flip and under a circular shift of $4W/8$ pixels for each model. The horizontal flip, a canonical semantics-preserving transformation (Wang et al., 2024b), yields a comparatively narrow distribution of absolute score differences ($|s_i - s_i^{flip}|$). In contrast, the circular shift produces a markedly wider spread of differences ($|s_i - s_i^{4W/8}|$), reflecting substantial instability. Consequently, we found insufficient evidence to reject the null hypothesis of non-stability.

**Does fine-tuning enhance the comprehension of 360-degree visual semantics?** To enhance CLIP models' comprehension of 360-degree visual semantics, we fine-tuned them using the approach described in Sec. 3. As shown in Fig. 4a, only the image encoder of the CLIP model is fine-tuned using LoRA. The LoRA rank is set to 8. The learning rate is fixed at $1 \times 10^{-5}$, and the batch size is set to 16. Fine-tuning is performed for 20 epochs using the AdamW optimizer (Loshchilov & Hutter, 2017) on a dataset derived from SUN360 (Xiao et al., 2012). More details on this fine-tuning dataset are provided in Sec. B.2. In addition, the procedure for determining the balancing parameter $\lambda$ based on knee-point detection is given in Sec. E.

Table 3 summarizes the outcomes of fine-tuning on ***360_real***. The values of $\lambda$ selected for ViT-B/32, ViT-B/16, and ViT-L/14 are 0.9889, 0.9899, and 0.9919, respectively. For all three fine-tuned models, the p-values for horizontal circular shifts at seven different magnitudes were consistently below the significance level ($\alpha = 0.01$). These results demonstrate that the fine-tuned CLIP models acquire a stable and robust understanding of 360-degree visual semantics, in contrast to their frozen counterparts. The results on ***360_syn*** using the fine-tuned CLIP models are presented in Table 12 (Sec. D), which further proves their improved comprehension of 360-degree visual semantics.

To further compare the semantic evaluation capability of frozen and fine-tuned models, we computed the CLIP scores ($s_i$) of all original 360-degree panoramic images in **360_real** using the frozen CLIP model (ViT-B/32) and its three fine-tuned variants. The resulting boxplots are shown in Fig. 7. As the balancing parameter $\lambda$ decreases from 1 to 0, the semantic evaluation capability of the fine-tuned CLIP model gradually recovers, demonstrating a clear trade-off between improved comprehension of 360-degree visual semantics and preservation of baseline semantic evaluation performance. When $\lambda = 0.9889$, the median CLIP score approaches that of the frozen model while retaining the enhanced understanding of 360-degree visual semantics, thereby validating our knee-point selection. Analogous results for ViT-B/16 and ViT-L/14 are presented in Fig. 12 (Sec. E).

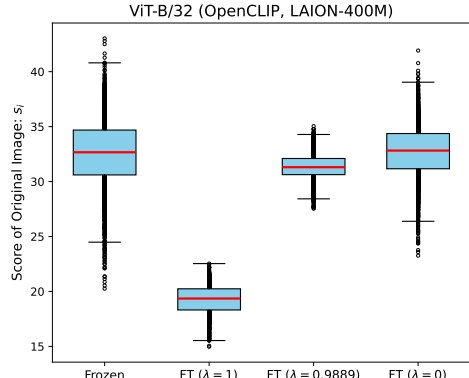

Figure 7: CLIP scores of original 360-degree panoramic images using a frozen CLIP model and its three fine-tuned (FT) versions.

## 5 RELATED WORK

**Contrastive Image-Text Learning.** Contrastive image-text models such as CLIP (Radford et al., 2021) learn joint embeddings of images and text prompts, enabling zero-shot classification and image-text retrieval. Subsequent works have focused on scalability and training efficiency (Fang et al., 2023; Sun et al., 2023; Tang et al., 2025; Tschannen et al., 2025; Xu et al., 2023; Zhai et al., 2023). However, these advances predominantly target perspective images. The capacity of standard CLIP models to capture the distinctive 360-degree visual semantics (arising from the complete spherical field of view) and 360-degree textual semantics (stemming from explicit format identifiers) inherent in panoramic image-text pairs remains largely untested. Although CLIP is widely employed as an evaluation metric for generated 360-degree content (Kalischek et al., 2025; Wang et al., 2024a; Zhang et al., 2024), its foundational comprehension of these panorama-specific semantic cues has not been systematically investigated. Our work addresses this gap by introducing statistical evaluation frameworks for probing CLIP on 360-degree panoramic image-text pairs.

**Text-Driven 360-Degree Panorama Generation.** Existing methods for text-driven 360-degree panorama generation can be broadly classified into two categories based on their input modalities: text-only generation and text-driven narrow field-of-view (NFoV) outpainting. Text-only generation approaches (Chen et al., 2022b; Feng et al., 2023; Wang et al., 2024a; Ye et al., 2024; Zhang et al., 2024) synthesize 360-degree panoramas through text prompts only. In contrast, text-driven NFoV outpainting methods (Kalischek et al., 2025; Lu et al., 2024; Wang et al., 2023; Zheng et al., 2025) use NFoV images alongside text prompts as inputs to generate complete 360-degree panoramas. For a detailed taxonomy and comprehensive review of these methodologies, we refer readers to the recent survey paper (Wang et al., 2025). In this study, we employ Diffusion360 (Feng et al., 2023), a state-of-the-art method for text-driven 360-degree panorama generation, to construct 360-degree panoramic image-text pairs for our investigation into CLIP models' capabilities.

## 6 DISCUSSION

**Conclusion.** We conducted the first systematic study of CLIP's understanding of 360-degree textual and visual semantics, evaluating multiple architectures across different training datasets. All models reliably exploit explicit panoramic format identifiers, confirming strong comprehension of 360-degree textual semantics and underscoring the need to include such cues in prompts for accurate evaluation. However, these models fail to maintain alignment under horizontal circular shifts, revealing a limited grasp of 360-degree visual semantics. To address this gap, we introduced a LoRA-based fine-tuning strategy that instills shift invariance and improves comprehension of 360-degree visual semantics, while incurring a slight degradation in baseline performance, highlighting the trade-off inherent in adapting CLIP for 360-degree panoramas. We will release the pre-trained LoRA weights for community use.

**Future Work.** Multimodal large language models (MLLMs) have demonstrated strong reasoning capabilities and promising performance across diverse multimodal tasks. Future work can investigate whether MLLMs may serve as evaluators for semantic alignment between generated 360-degree panoramic images and their corresponding textual prompts, potentially offering complementary perspectives beyond CLIP-based metrics.

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

## A    THE USE OF LARGE LANGUAGE MODELS (LLMs)

The authors used OpenAI's ChatGPT for two minor purposes only:

1. Removing the panoramic format identifiers of base prompts generated from BLIP-2 (Li et al., 2023), as detailed in Sec. 4.1;

2. Assisting with grammar, wording, general writing polish.

# B MORE IMPLEMENTATION DETAILS

## B.1 FLOWCHART OF DATASET GENERATION

Fig. 8 shows the flowchart to produce the two paired image-text datasets (*360_real* and *360_syn*) used in our evaluation experiments.

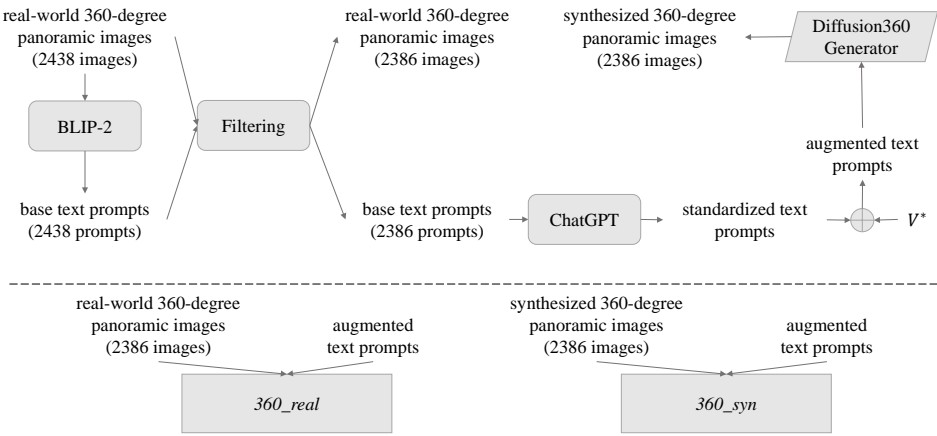

Figure 8: Diagram to show the generation process of paired image-text datasets. The format cue $V^*$ is "*<360panorama>,* ".

## B.2 FINE-TUNING DATASET

The fine-tuning dataset is derived from SUN360 (Xiao et al., 2012). Specifically, we adopt the processed version introduced by Zheng et al. (2025), which contains 25,000 360-degree panoramic image-text pairs with standardized text prompts generated from BLIP-2 (Li et al., 2023). Following the same procedure illustrated in Fig. 8, we filter out text prompts containing directional cues (e.g.,"*in the middle*"), yielding 23,811 pairs. Each remaining prompt is prepended with the string "*<360panorama>,* ". The resulting 23,811 augmented prompts, paired with their corresponding 360-degree panoramic images, constituted the fine-tuning dataset.

# C ADDITIONAL RESULTS OF 360-DEGREE TEXTUAL SEMANTICS

## C.1 NORMALITY TEST FOR 360-DEGREE TEXTUAL SEMANTICS

The standard choice for paired data, the paired t-test, assumes that the differences between pairs are normally distributed. We formally checked this assumption for these score differences ($s - s^u$) using the Shapiro-Wilk test (Shapiro & Wilk, 1965). The results, presented in Table 4, show that most p-values were below the standard significance level ($\alpha = 0.01$), which means that there is significant evidence to reject the null hypothesis of normality for these score differences. To keep consistency, the non-parametric Wilcoxon Signed-Rank test was employed.

Table 4: **[OpenCLIP, LAION-400M]** [$V^*$ ="*<360panorama>, *"], the Shapiro-Wilk test (Shapiro & Wilk, 1965) results for different CLIP models on the two paired image-text datasets, where the null hypothesis is the distribution of the score differences ($s - s^u$) is normal, and the significance level ($\alpha$) is 0.01. The p-values less than $\alpha$ are in bold.

| ViT | $U^*$ ="" | | | | $U^*$ ="*image, *" | | | |
| | *360_real* | | *360_syn* | | *360_real* | | *360_syn* | |
| | *statistic* | *p-value* | *statistic* | *p-value* | *statistic* | *p-value* | *statistic* | *p-value* |
|---|---|---|---|---|---|---|---|---|
| B/32 | 0.9979 | **0.0035** | 0.9957 | **0** | 0.9981 | **0.0068** | 0.9967 | **0.0001** |
| B/16 | 0.9980 | **0.0050** | 0.9342 | **0** | 0.9991 | 0.3060 | 0.9129 | **0** |
| L/14 | 0.9935 | **0** | 0.9982 | **0.0099** | 0.9940 | **0** | 0.9977 | **0.0014** |

## C.2 OTHER GENERIC CUES

Here, the specific format cue $V^*$ in the original prompts was replaced by the other two distinct generic cues ($U^*$) to produce the generic prompts $T^u$ and their corresponding CLIP scores $s^u$. These scores were then compared against the original scores $s$. The results of these analyses, listed in Table 5, consistently demonstrated that for all tested CLIP models and across all generic keyword substitutions on both 360-degree panoramic paired datasets, the null hypothesis was rejected ($p < 0.01$ for all cases). This outcome offers robust statistical evidence that the evaluated CLIP models effectively discern and leverage 360-degree panorama-specific textual cues, leading to significantly enhanced image-text alignment when such specific cues are present.

Table 5: **[OpenCLIP, LAION-400M]** [$V^*$ ="*<360panorama>, *"], the Wilcoxon Signed-Rank test (Wilcoxon, 1945) results for different CLIP models on the two paired image-text datasets (*360_real* and *360_syn*), where the null hypothesis is the original score $s$ is not greater than the generic score $s^u$, and the significance level ($\alpha$) is 0.01. The p-values less than $\alpha$ are in bold.

| ViT | $U^*$ ="*photo, *" | | | | $U^*$ ="*picture, *" | | | |
| | *360_real* | | *360_syn* | | *360_real* | | *360_syn* | |
| | *statistic* | *p-value* | *statistic* | *p-value* | *statistic* | *p-value* | *statistic* | *p-value* |
|---|---|---|---|---|---|---|---|---|
| B/32 | 2847690 | **0** | 2847217 | **0** | 2847688 | **0** | 2847202 | **0** |
| B/16 | 2847553 | **0** | 2700137 | **0** | 2847525 | **0** | 2654132 | **0** |
| L/14 | 2847690 | **0** | 2847501 | **0** | 2847691 | **0** | 2847482 | **0** |

## C.3 OTHER FORMAT CUES

Building upon the evaluation of 360-degree textual semantics through keyword manipulation (Sec. 4.2), which used the format cue $V^*$ ="*<360panorama>, *", we investigated the impact of alternative cues. Specifically, we tested "*a 360 degree view of *" and "*360 photo, *", both explicitly indicating the 360-degree panoramic format. The one-tailed Wilcoxon test results for these cues

are presented in Tables 6-7 and Tables 8-9, respectively. Across all tested CLIP models and generic keyword substitutions on both 360-degree panoramic paired datasets, the null hypothesis was consistently rejected ($p < 0.01$ for all cases). These findings further provide strong statistical evidence that the evaluated CLIP models effectively discern and utilize 360-degree panorama-specific textual cues, leading to significantly higher image-text alignment when such cues are present.

Table 6: **[OpenCLIP, LAION-400M]** [$V^*$ =“*a 360 degree view of* ”], the Wilcoxon Signed-Rank test (Wilcoxon, 1945) results for different CLIP models on the two paired image-text datasets, where the null hypothesis is the original score $s$ is not greater than the generic score $s^u$, and the significance level ($\alpha$) is 0.01. The p-values less than $\alpha$ are in bold.

| ViT | $U^*$ =“” | | | | $U^*$ =“*image,* ” | | | |
| | *360_real* | | *360_syn* | | *360_real* | | *360_syn* | |
| | *statistic* | *p-value* | *statistic* | *p-value* | *statistic* | *p-value* | *statistic* | *p-value* |
|---|---|---|---|---|---|---|---|---|
| B/32 | 2847669 | **0** | 2847084 | **0** | 2847677 | **0** | 2844782 | **0** |
| B/16 | 2847687 | **0** | 2844426 | **0** | 2847607 | **0** | 2779991 | **0** |
| L/14 | 2847690 | **0** | 2847572 | **0** | 2847662 | **0** | 2847145 | **0** |

Table 7: **[OpenCLIP, LAION-400M]** [$V^*$ =“*a 360 degree view of* ”]. The rest caption is as for Table 6.

| ViT | $U^*$ =“*photo,* ” | | | | $U^*$ =“*picture,* ” | | | |
| | *360_real* | | *360_syn* | | *360_real* | | *360_syn* | |
| | *statistic* | *p-value* | *statistic* | *p-value* | *statistic* | *p-value* | *statistic* | *p-value* |
|---|---|---|---|---|---|---|---|---|
| B/32 | 2847685 | **0** | 2846656 | **0** | 2847685 | **0** | 2846657 | **0** |
| B/16 | 2847690 | **0** | 2841828 | **0** | 2847687 | **0** | 2842667 | **0** |
| L/14 | 2847691 | **0** | 2847520 | **0** | 2847690 | **0** | 2847442 | **0** |

Table 8: **[OpenCLIP, LAION-400M]** [$V^*$ =“*360 photo,* ”]. The rest caption is as for Table 6.

| ViT | $U^*$ =“” | | | | $U^*$ =“*image,* ” | | | |
| | *360_real* | | *360_syn* | | *360_real* | | *360_syn* | |
| | *statistic* | *p-value* | *statistic* | *p-value* | *statistic* | *p-value* | *statistic* | *p-value* |
|---|---|---|---|---|---|---|---|---|
| B/32 | 2847626 | **0** | 2847345 | **0** | 2847438 | **0** | 2844193 | **0** |
| B/16 | 2847658 | **0** | 2808619 | **0** | 2846456 | **0** | 2561918 | **0** |
| L/14 | 2847682 | **0** | 2847230 | **0** | 2847637 | **0** | 2846301 | **0** |

Table 9: **[OpenCLIP, LAION-400M]** [$V^* =$"*360 photo,* "]. The rest caption is as for Table 6.

| ViT | $U^* =$"*photo,* " | | | | $U^* =$"*picture,* " | | | |
| --- | --- | --- | --- | --- | --- | --- | --- | --- |
| | *360_real* | | *360_syn* | | *360_real* | | *360_syn* | |
| | *statistic* | *p-value* | *statistic* | *p-value* | *statistic* | *p-value* | *statistic* | *p-value* |
| B/32 | 2847687 | **0** | 2847433 | **0** | 2847659 | **0** | 2847200 | **0** |
| B/16 | 2847656 | **0** | 2828690 | **0** | 2847608 | **0** | 2817956 | **0** |
| L/14 | 2847691 | **0** | 2847438 | **0** | 2847687 | **0** | 2847159 | **0** |

## D  ADDITIONAL RESULTS OF 360-DEGREE VISUAL SEMANTICS

To justify the choice of the Wilcoxon Signed-Rank test (Wilcoxon, 1945) for evaluating the null hypothesis that $|s - s^{\delta_j}|$ is statistically greater than or equal to the stability bound $\beta$, we assessed the normality of the absolute score differences ($|s - s^{\delta_j}|$) between the original and shifted CLIP scores using the Shapiro-Wilk test (Shapiro & Wilk, 1965). The results, detailed in Table 10, indicate that for all datasets, the p-values were below a commonly used significance level ($\alpha = 0.01$). Consequently, the null hypothesis of normality for the absolute differences was rejected. This finding validates the use of the non-parametric Wilcoxon Signed-Rank test for our analyses.

Table 10: **[OpenCLIP, LAION-400M]**, the Shapiro-Wilk test (Shapiro & Wilk, 1965) results under horizontal circular shift of various $\delta_j$ pixels for different CLIP models on the ***360_real*** dataset, where the null hypothesis is the distribution of the absolute differences ($|s - s^{\delta_j}|$) is not significantly different from a normal distribution and the significance level ($\alpha$) is 0.01. The p-values less than $\alpha$ are in bold.

| ViT | $\delta_j$ | $W/8$ | $2W/8$ | $3W/8$ | $4W/8$ | $5W/8$ | $6W/8$ | $7W/8$ |
|---|---|---|---|---|---|---|---|---|
| B/32 | *statistic\|p-value* | 0.8715\|**0** | 0.8866\|**0** | 0.8898\|**0** | 0.8737\|**0** | 0.8727\|**0** | 0.8624\|**0** | 0.8923\|**0** |
| B/16 | *statistic\|p-value* | 0.8630\|**0** | 0.8688\|**0** | 0.8667\|**0** | 0.8636\|**0** | 0.8598\|**0** | 0.8694\|**0** | 0.8759\|**0** |
| L/14 | *statistic\|p-value* | 0.8404\|**0** | 0.8559\|**0** | 0.8657\|**0** | 0.8694\|**0** | 0.8654\|**0** | 0.8588\|**0** | 0.8419\|**0** |

Table 11 and Table 12 present the results of the Wilcoxon Signed-Rank tests on the ***360_syn*** dataset using frozen and fine-tuned CLIP models, respectively.

Table 11: **[OpenCLIP, LAION-400M]**, the Wilcoxon Signed-Rank test (Wilcoxon, 1945) results under horizontal circular shift of various $\delta_j$ pixels for different CLIP models on the ***360_syn*** dataset, where the null hypothesis ($H_0$) is that $|s - s^{\delta_j}|$ is greater than or equal to the stability bound $\beta$, and the significance level ($\alpha$) is 0.01. The p-values less than $\alpha$ are in bold.

| ViT | $\beta$ | $\delta_j$ | $W/8$ | $2W/8$ | $3W/8$ | $4W/8$ | $5W/8$ | $6W/8$ | $7W/8$ |
|---|---|---|---|---|---|---|---|---|---|
| B/32 | 1.7096 | *statistic* | 1028188 | 1496678 | 1694678 | 1704520 | 1645991 | 1492581 | 1054393 |
| | | *p-value* | **0** | 0.9848 | 1 | 1 | 1 | 0.9794 | **0** |
| B/16 | 1.5901 | *statistic* | 1223380 | 1696387 | 1834997 | 1887851 | 1851771 | 1725702 | 1225625 |
| | | *p-value* | **0** | 1 | 1 | 1 | 1 | 1 | **0** |
| L/14 | 1.4677 | *statistic* | 1373467 | 1931962 | 2062222 | 2122936 | 2135268 | 1945826 | 1387542 |
| | | *p-value* | 0.0672 | 1 | 1 | 1 | 1 | 1 | 0.1404 |

Table 12: **[Fine-Tuned, OpenCLIP, LAION-400M]**. The rest caption is as for Table 11.

| $\lambda$ | ViT | $\beta$ | $\delta_j$ | $W/8$ | $2W/8$ | $3W/8$ | $4W/8$ | $5W/8$ | $6W/8$ | $7W/8$ |
|---|---|---|---|---|---|---|---|---|---|---|
| 0.9889 | B/32 | 1.7096 | *statistic* | 0 | 0 | 0 | 0 | 0 | 0 | 0 |
| | | | *p-value* | **0** | **0** | **0** | **0** | **0** | **0** | **0** |
| 0.9899 | B/16 | 1.5901 | *statistic* | 0 | 0 | 0 | 0 | 0 | 0 | 0 |
| | | | *p-value* | **0** | **0** | **0** | **0** | **0** | **0** | **0** |
| 0.9919 | L/14 | 1.4677 | *statistic* | 0 | 0 | 0 | 0 | 0 | 0 | 0 |
| | | | *p-value* | **0** | **0** | **0** | **0** | **0** | **0** | **0** |

# E  DETERMINATION OF $\lambda$

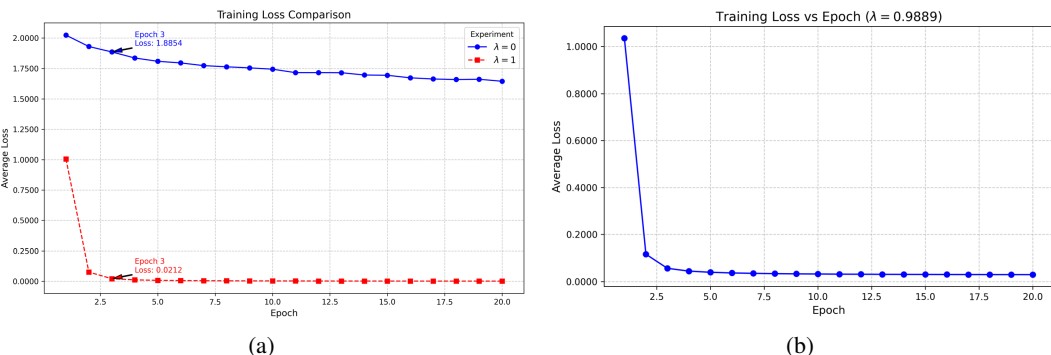

(a)                                                    (b)

Figure 9: Fine-tuning loss curves using different $\lambda$ values of ViT-B/32 (OpenCLIP, LAION-400M).

To determine an appropriate value of $\lambda$ for weighting the two components in $L_{FT}$, we adopt a data-driven approach based on knee-point detection. Specifically, we first set $\lambda = 1$, record the corresponding loss curve, and detect its knee point using the kneed Python package.[1] The loss value at this knee point is denoted as $l_1^{knee}$. Next, we set $\lambda = 0$, obtain the corresponding loss curve, and record the loss value at the same knee point detected under $\lambda = 1$, denoted as $l_0^{knee}$. To balance the contributions of the two components, we require the weighted losses to be equal at this knee point: $\lambda \cdot l_1^{knee} = (1 - \lambda) \cdot l_0^{knee}$. Solving for $\lambda$ yields:

$$\lambda = \frac{l_0^{knee}}{(l_0^{knee} + l_1^{knee})} \ . \tag{7}$$

As shown in Fig. 9a, the knee point occurs at Epoch 3. The corresponding loss values of $l_1^{knee}$ and $l_0^{knee}$ are 0.0212 and 1.8854, respectively. According to the Eq. (7), the balancing parameter $\lambda$ is 0.9889, and the resulting loss curve is illustrated in Fig. 9b.

Fig. 9 demonstrates the full set of loss curves used to determine $\lambda$ for ViT-B/32. Analogous results for ViT-B/16 and ViT-L/14 are provided in Fig. 10 and Fig. 11, respectively. In addition, we demonstrate the qualitative comparisons of semantic evaluation performance between the frozen and fine-tuned models for these two configurations in Fig. 12.

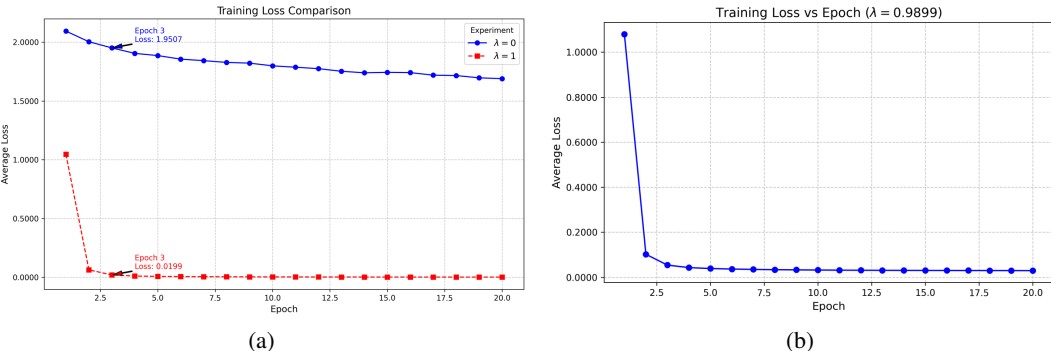

(a)                                                    (b)

Figure 10: Fine-tuning loss curves using different $\lambda$ values of ViT-B/16 (OpenCLIP, LAION-400M).

---

[1]https://kneed.readthedocs.io/en/stable/

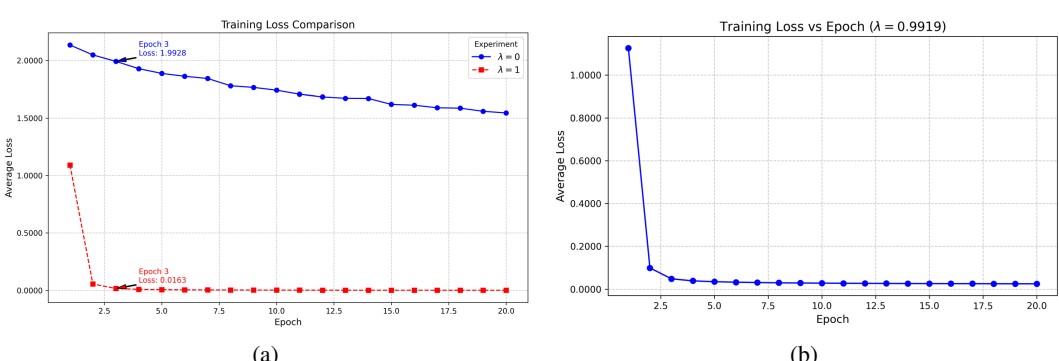

Figure 11: Fine-tuning loss curves using different $\lambda$ values of ViT-L/14 (OpenCLIP, LAION-400M).

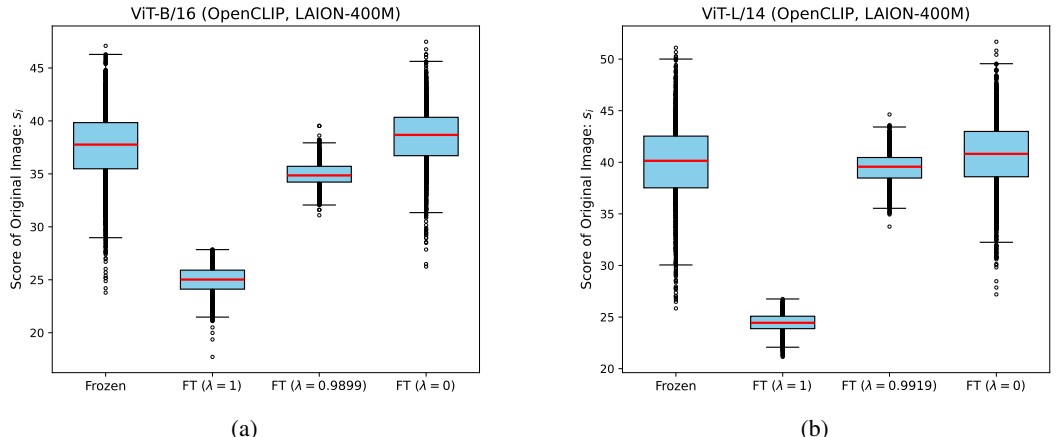

Figure 12: CLIP scores of original 360-degree panoramic images using a frozen CLIP model and its three fine-tuned (FT) versions.

# F RESULTS OF CLIP MODELS TRAINED ON LAION-2B

## F.1 RESULTS OF 360-DEGREE TEXTUAL SEMANTICS

Table 13: **[OpenCLIP, LAION-2B]** [$V^* =$"<360panorama>, "], the Wilcoxon Signed-Rank test (Wilcoxon, 1945) results for different CLIP models on the two paired image-text datasets (*360_real* and *360_syn*), where the null hypothesis is the original score $s$ is not greater than the generic score $s^u$, and the significance level ($\alpha$) is 0.01. The p-values less than $\alpha$ are in bold.

| ViT | $U^* =$"" | | | | $U^* =$"image, " | | | |
|---|---|---|---|---|---|---|---|---|
| | 360_real | | 360_syn | | 360_real | | 360_syn | |
| | statistic | p-value | statistic | p-value | statistic | p-value | statistic | p-value |
| B/32 | 2847683 | **0** | 2846939 | **0** | 2847679 | **0** | 2845716 | **0** |
| B/16 | 2847523 | **0** | 2839923 | **0** | 2847206 | **0** | 2809215 | **0** |
| L/14 | 2847643 | **0** | 2843198 | **0** | 2847568 | **0** | 2835474 | **0** |

Table 14: **[OpenCLIP, LAION-2B]** [$V^* =$"<360panorama>, "]. The rest caption is as for Table 13.

| ViT | $U^* =$"photo, " | | | | $U^* =$"picture, " | | | |
|---|---|---|---|---|---|---|---|---|
| | 360_real | | 360_syn | | 360_real | | 360_syn | |
| | statistic | p-value | statistic | p-value | statistic | p-value | statistic | p-value |
| B/32 | 2847684 | **0** | 2846959 | **0** | 2847685 | **0** | 2846961 | **0** |
| B/16 | 2847525 | **0** | 2839250 | **0** | 2847518 | **0** | 2840563 | **0** |
| L/14 | 2847636 | **0** | 2843751 | **0** | 2847651 | **0** | 2839435 | **0** |

## F.2 RESULTS OF 360-DEGREE VISUAL SEMANTICS

Table 15: **[OpenCLIP, LAION-2B]**, the Wilcoxon Signed-Rank test (Wilcoxon, 1945) results under horizontal circular shift of various $\delta_j$ pixels for different CLIP models on the **360_real** dataset, where the null hypothesis ($H_0$) is that $|s - s^{\delta_j}|$ is greater than or equal to the stability bound $\beta$, and the significance level ($\alpha$) is 0.01. The p-values less than $\alpha$ are in bold.

| ViT | $\beta$ | $\delta_j$ | $W/8$ | $2W/8$ | $3W/8$ | $4W/8$ | $5W/8$ | $6W/8$ | $7W/8$ |
|---|---|---|---|---|---|---|---|---|---|
| B/32 | 1.5895 | statistic | 824700 | 1462504 | 1644768 | 1732685 | 1669792 | 1414524 | 798663 |
| | | p-value | **0** | 0.8747 | 1 | 1 | 1 | 0.3909 | **0** |
| B/16 | 1.4789 | statistic | 950472 | 1437802 | 1710235 | 1751568 | 1683293 | 1423853 | 950834 |
| | | p-value | **0** | 0.6608 | 1 | 1 | 1 | 0.5001 | **0** |
| L/14 | 1.3514 | statistic | 911944 | 1621999 | 1894766 | 2023043 | 1904321 | 1567941 | 826992 |
| | | p-value | **0** | 1 | 1 | 1 | 1 | 1 | **0** |

Table 16: **[Fine-Tuned, OpenCLIP, LAION-2B]**. The rest caption is as for Table 15.

| $\lambda$ | ViT | $\beta$ | $\delta_j$ | $W/8$ | $2W/8$ | $3W/8$ | $4W/8$ | $5W/8$ | $6W/8$ | $7W/8$ |
|---|---|---|---|---|---|---|---|---|---|---|
| 0.9701 | B/32 | 1.5895 | *statistic* | 0 | 0 | 0 | 0 | 0 | 0 | 0 |
| | | | *p-value* | **0** | **0** | **0** | **0** | **0** | **0** | **0** |
| 0.9849 | B/16 | 1.4789 | *statistic* | 0 | 0 | 0 | 0 | 0 | 0 | 0 |
| | | | *p-value* | **0** | **0** | **0** | **0** | **0** | **0** | **0** |
| 0.9922 | L/14 | 1.3514 | *statistic* | 0 | 0 | 0 | 0 | 0 | 0 | 0 |
| | | | *p-value* | **0** | **0** | **0** | **0** | **0** | **0** | **0** |

Table 17: **[OpenCLIP, LAION-2B]**, the Wilcoxon Signed-Rank test (Wilcoxon, 1945) results under horizontal circular shift of various $\delta_j$ pixels for different CLIP models on the ***360_syn*** dataset, where the null hypothesis ($H_0$) is that $|s - s^{\delta_j}|$ is greater than or equal to the stability bound $\beta$, and the significance level ($\alpha$) is 0.01. The p-values less than $\alpha$ are in bold.

| ViT | $\beta$ | $\delta_j$ | $W/8$ | $2W/8$ | $3W/8$ | $4W/8$ | $5W/8$ | $6W/8$ | $7W/8$ |
|---|---|---|---|---|---|---|---|---|---|
| B/32 | 1.7699 | *statistic* | 844832 | 1258762 | 1500098 | 1526052 | 1481424 | 1298442 | 808795 |
| | | *p-value* | **0** | **0** | 0.9883 | 0.9988 | 0.9564 | **0.0001** | **0** |
| B/16 | 1.8719 | *statistic* | 970833 | 1355894 | 1554592 | 1543064 | 1615806 | 1315390 | 887415 |
| | | *p-value* | **0** | 0.0217 | 0.9999 | 0.9998 | 1 | **0.0006** | **0** |
| L/14 | 1.6056 | *statistic* | 788979 | 1349352 | 1597760 | 1688313 | 1647770 | 1379937 | 795128 |
| | | *p-value* | **0** | 0.0134 | 1 | 1 | 1 | 0.0960 | **0** |

Table 18: **[Fine-Tuned, OpenCLIP, LAION-2B]**. The rest caption is as for Table 17.

| $\lambda$ | ViT | $\beta$ | $\delta_j$ | $W/8$ | $2W/8$ | $3W/8$ | $4W/8$ | $5W/8$ | $6W/8$ | $7W/8$ |
|---|---|---|---|---|---|---|---|---|---|---|
| 0.9701 | B/32 | 1.7699 | *statistic* | 0 | 0 | 0 | 0 | 0 | 0 | 0 |
| | | | *p-value* | **0** | **0** | **0** | **0** | **0** | **0** | **0** |
| 0.9849 | B/16 | 1.8719 | *statistic* | 0 | 0 | 0 | 0 | 0 | 0 | 0 |
| | | | *p-value* | **0** | **0** | **0** | **0** | **0** | **0** | **0** |
| 0.9922 | L/14 | 1.6056 | *statistic* | 0 | 0 | 0 | 0 | 0 | 0 | 0 |
| | | | *p-value* | **0** | **0** | **0** | **0** | **0** | **0** | **0** |

# G    RESULTS OF CLIP MODELS FROM OPENAI

## G.1    RESULTS OF 360-DEGREE TEXTUAL SEMANTICS

Table 19: **[OpenAI]** [$V^* =$"*<360panorama>, *"], the Wilcoxon Signed-Rank test (Wilcoxon, 1945) results for different CLIP models on the two paired image-text datasets (*360_real* and *360_syn*), where the null hypothesis is the original score $s$ is not greater than the generic score $s^u$, and the significance level ($\alpha$) is 0.01. The p-values less than $\alpha$ are in bold.

| ViT | $U^* =$"" | | | | $U^* =$"*image, *" | | | |
| | *360_real* | | *360_syn* | | *360_real* | | *360_syn* | |
| | *statistic* | *p-value* | *statistic* | *p-value* | *statistic* | *p-value* | *statistic* | *p-value* |
|---|---|---|---|---|---|---|---|---|
| B/32 | 2847642 | **0** | 2843908 | **0** | 2847557 | **0** | 2812782 | **0** |
| B/16 | 2847591 | **0** | 2841449 | **0** | 2847530 | **0** | 2750650 | **0** |
| L/14 | 2847676 | **0** | 2847226 | **0** | 2847648 | **0** | 2846129 | **0** |

Table 20: **[OpenAI]** [$V^* =$"*<360panorama>, *"]. The rest caption is as for Table 19.

| ViT | $U^* =$"*photo, *" | | | | $U^* =$"*picture, *" | | | |
| | *360_real* | | *360_syn* | | *360_real* | | *360_syn* | |
| | *statistic* | *p-value* | *statistic* | *p-value* | *statistic* | *p-value* | *statistic* | *p-value* |
|---|---|---|---|---|---|---|---|---|
| B/32 | 2847521 | **0** | 2814142 | **0** | 2847569 | **0** | 2812673 | **0** |
| B/16 | 2847556 | **0** | 2793487 | **0** | 2847601 | **0** | 2785342 | **0** |
| L/14 | 2847580 | **0** | 2846677 | **0** | 2847676 | **0** | 2846846 | **0** |

## G.2    RESULTS OF 360-DEGREE VISUAL SEMANTICS

Table 21: **[OpenAI]**, the Wilcoxon Signed-Rank test (Wilcoxon, 1945) results under horizontal circular shift of various $\delta_j$ pixels for different CLIP models on the **360_real** dataset, where the null hypothesis ($H_0$) is that $|s - s^{\delta_j}|$ is greater than or equal to the stability bound $\beta$, and the significance level ($\alpha$) is 0.01. The p-values less than $\alpha$ are in bold.

| ViT | $\beta$ | $\delta_j$ | $W/8$ | $2W/8$ | $3W/8$ | $4W/8$ | $5W/8$ | $6W/8$ | $7W/8$ |
|---|---|---|---|---|---|---|---|---|---|
| B/32 | 1.0703 | *statistic* | 766434 | 1346792 | 1492278 | 1593231 | 1459658 | 1242843 | 721201 |
| | | *p-value* | **0** | 0.0110 | 0.9790 | 1 | 0.8564 | **0** | **0** |
| B/16 | 0.8607 | *statistic* | 833253 | 1323362 | 1633531 | 1813687 | 1667101 | 1398948 | 890113 |
| | | *p-value* | **0** | **0.0014** | 1 | 1 | 1 | 0.2297 | **0** |
| L/14 | 1.0147 | *statistic* | 562349 | 1241492 | 1540685 | 1646453 | 1587058 | 1285634 | 577713 |
| | | *p-value* | **0** | **0** | 0.9997 | 1 | 1 | **0** | **0** |

Table 22: **[Fine-Tuned, OpenAI]**. The rest caption is as for Table 21.

| $\lambda$ | ViT | $\beta$ | $\delta_j$ | $W/8$ | $2W/8$ | $3W/8$ | $4W/8$ | $5W/8$ | $6W/8$ | $7W/8$ |
|---|---|---|---|---|---|---|---|---|---|---|
| 0.9831 | B/32 | 1.0703 | *statistic* | 0 | 0 | 0 | 0 | 0 | 0 | 0 |
| | | | *p-value* | **0** | **0** | **0** | **0** | **0** | **0** | **0** |
| 0.9839 | B/16 | 0.8607 | *statistic* | 0 | 0 | 0 | 0 | 0 | 0 | 0 |
| | | | *p-value* | **0** | **0** | **0** | **0** | **0** | **0** | **0** |
| 0.9882 | L/14 | 1.0147 | *statistic* | 0 | 0 | 0 | 0 | 0 | 0 | 0 |
| | | | *p-value* | **0** | **0** | **0** | **0** | **0** | **0** | **0** |

Table 23: **[OpenAI]**, the Wilcoxon Signed-Rank test (Wilcoxon, 1945) results under horizontal circular shift of various $\delta_j$ pixels for different CLIP models on the ***360_syn*** dataset, where the null hypothesis ($H_0$) is that $|s - s^{\delta_j}|$ is greater than or equal to the stability bound $\beta$, and the significance level ($\alpha$) is 0.01. The p-values less than $\alpha$ are in bold.

| ViT | $\beta$ | $\delta_j$ | $W/8$ | $2W/8$ | $3W/8$ | $4W/8$ | $5W/8$ | $6W/8$ | $7W/8$ |
|---|---|---|---|---|---|---|---|---|---|
| B/32 | 1.0822 | *statistic* | 792795 | 1158722 | 1387464 | 1512676 | 1439411 | 1226799 | 798591 |
| | | *p-value* | **0** | **0** | 0.1398 | 0.9958 | 0.6781 | **0** | **0** |
| B/16 | 1.0704 | *statistic* | 774810 | 1147851 | 1489787 | 1551690 | 1504444 | 1189310 | 749265 |
| | | *p-value* | **0** | **0** | 0.9750 | 0.9999 | 0.9917 | **0** | **0** |
| L/14 | 1.1995 | *statistic* | 565237 | 1058911 | 1355776 | 1450019 | 1373690 | 1077157 | 545371 |
| | | *p-value* | **0** | **0** | 0.0216 | 0.7816 | 0.0681 | **0** | **0** |

Table 24: **[Fine-Tuned, OpenAI]**. The rest caption is as for Table 23.

| $\lambda$ | ViT | $\beta$ | $\delta_j$ | $W/8$ | $2W/8$ | $3W/8$ | $4W/8$ | $5W/8$ | $6W/8$ | $7W/8$ |
|---|---|---|---|---|---|---|---|---|---|---|
| 0.9831 | B/32 | 1.0822 | *statistic* | 0 | 0 | 0 | 0 | 0 | 0 | 0 |
| | | | *p-value* | **0** | **0** | **0** | **0** | **0** | **0** | **0** |
| 0.9839 | B/16 | 1.0704 | *statistic* | 0 | 0 | 0 | 0 | 0 | 0 | 0 |
| | | | *p-value* | **0** | **0** | **0** | **0** | **0** | **0** | **0** |
| 0.9882 | L/14 | 1.1995 | *statistic* | 0 | 0 | 0 | 0 | 0 | 0 | 0 |
| | | | *p-value* | **0** | **0** | **0** | **0** | **0** | **0** | **0** |

# H    GENERALIZATION CAPABILITY OF FINE-TUNED MODELS

## H.1    GENERALIZATION TO NATURAL LANDSCAPES

Table 25: **[Fine-Tuned, OpenCLIP, LAION-400M]**, the Wilcoxon Signed-Rank test results under horizontal circular shift of various $\delta_j$ pixels for different CLIP models on the ***360_nature*** dataset, where the null hypothesis is that $|s - s^{\delta_j}|$ is greater than or equal to the stability bound $\beta$, and the significance level ($\alpha$) is 0.01. The p-values less than $\alpha$ are in bold.

| $\lambda$ | ViT | $\beta$ | $\delta_j$ | $W/8$ | $2W/8$ | $3W/8$ | $4W/8$ | $5W/8$ | $6W/8$ | $7W/8$ |
|---|---|---|---|---|---|---|---|---|---|---|
| 0.9889 | B/32 | 1.7401 | *statistic* | 0 | 0 | 0 | 0 | 0 | 0 | 0 |
| | | | *p-value* | **0** | **0** | **0** | **0** | **0** | **0** | **0** |
| 0.9899 | B/16 | 1.9949 | *statistic* | 0 | 0 | 0 | 0 | 0 | 0 | 0 |
| | | | *p-value* | **0** | **0** | **0** | **0** | **0** | **0** | **0** |
| 0.9919 | L/14 | 1.7896 | *statistic* | 0 | 0 | 0 | 0 | 0 | 0 | 0 |
| | | | *p-value* | **0** | **0** | **0** | **0** | **0** | **0** | **0** |

To test the generalization capability of fine-tuned models to other scenes, we utilized Diffusion360 (Feng et al., 2023) to generate 500 360-degree panoramas of natural landscapes, with text prompts produced by ChatGPT. We refer to this new dataset as ***360_nature***.

Following the procedure in Sec. 2.2, the stability bounds $\beta$ on the ***360_nature*** dataset for ViT-B/32, ViT-B/16, and ViT-L/14 are 1.7401, 1.9949, and 1.7896, respectively. Using these stability bounds, we conducted one-sided Wilcoxon Signed-Rank test for CLIP models trained on LAION-400M. As reported in Table 25, all p-values were consistently below the significance level ($\alpha = 0.01$), indicating that the fine-tuned CLIP models continue to exhibit a robust understanding of 360-degree visual semantics on natural-landscape scenes. These results demonstrate the strong generalization capability of our fine-tuned models to more diverse scene types.

## H.2    GENERALIZATION TO UNSEEN SHIFT MAGNITUDES

Table 26: **[Fine-Tuned, OpenCLIP, LAION-400M]**, the Wilcoxon Signed-Rank test results under horizontal circular shift of various $\delta_j$ pixels for different CLIP models on the ***360_real*** dataset, where the null hypothesis is that $|s - s^{\delta_j}|$ is greater than or equal to the stability bound $\beta$, and the significance level ($\alpha$) is 0.01. The p-values less than $\alpha$ are in bold.

| $\lambda$ | ViT | $\beta$ | $\delta_j$ | 110 | 210 | 310 | 410 | 510 | 610 | 710 |
|---|---|---|---|---|---|---|---|---|---|---|
| 0.9889 | B/32 | 1.7919 | *statistic* | 0 | 0 | 0 | 0 | 0 | 0 | 0 |
| | | | *p-value* | **0** | **0** | **0** | **0** | **0** | **0** | **0** |
| 0.9899 | B/16 | 1.6547 | *statistic* | 0 | 0 | 0 | 0 | 0 | 0 | 0 |
| | | | *p-value* | **0** | **0** | **0** | **0** | **0** | **0** | **0** |
| 0.9919 | L/14 | 1.4245 | *statistic* | 0 | 0 | 0 | 0 | 0 | 0 | 0 |
| | | | *p-value* | **0** | **0** | **0** | **0** | **0** | **0** | **0** |

To evaluate generalization to unseen shift magnitudes, we modified the training procedure so that the shift distance $\Delta$ was randomly selected from $\{0, 32, 64, \ldots, 992\}$. We then carried out the Wilcoxon Signed-Rank test under horizontal circular shift of 110, 210, 310, 410, 510, 610, and 710 pixels on both the ***360_real*** and ***360_syn*** datasets. As shown in Table 26 and Table 27, all p-values at these seven unseen shift magnitudes were consistently below the significance level ($\alpha = 0.01$), demonstrating that these fine-tuned models still have a robust understanding of 360-degree visual semantics. These results reflect a strong generalization capability of the fine-tuned model.

Table 27: **[Fine-Tuned, OpenCLIP, LAION-400M]**, the Wilcoxon Signed-Rank test results under horizontal circular shift of various $\delta_j$ pixels for different CLIP models on the ***360_syn*** dataset, where the null hypothesis is that $|s - s^{\delta_j}|$ is greater than or equal to the stability bound $\beta$, and the significance level ($\alpha$) is 0.01. The p-values less than $\alpha$ are in bold.

| $\lambda$ | ViT | $\beta$ | $\delta_j$ | 110 | 210 | 310 | 410 | 510 | 610 | 710 |
|---|---|---|---|---|---|---|---|---|---|---|
| 0.9889 | B/32 | 1.7096 | *statistic* | 0 | 0 | 0 | 0 | 0 | 0 | 0 |
| | | | *p-value* | **0** | **0** | **0** | **0** | **0** | **0** | **0** |
| 0.9899 | B/16 | 1.5901 | *statistic* | 0 | 0 | 0 | 0 | 0 | 0 | 0 |
| | | | *p-value* | **0** | **0** | **0** | **0** | **0** | **0** | **0** |
| 0.9919 | L/14 | 1.4677 | *statistic* | 0 | 0 | 0 | 0 | 0 | 0 | 0 |
| | | | *p-value* | **0** | **0** | **0** | **0** | **0** | **0** | **0** |

# I COMPARISON OF DIFFERENT FINE-TUNING METHODS

To investigate the impact of different fine-tuning strategies on the fine-tuned CLIP model's comprehension of 360-degree visual semantics, we extended our analysis beyond the LoRA-based image-encoder tuning used in the main paper. Specifically, we examined two additional configurations: (1) applying LoRA to both the image and text encoders, and (2) full fine-tuning of the image encoder without LoRA.

The Wilcoxon Signed-Rank test results for LoRA fine-tuning on both encoders and full fine-tuning on the image encoder are summarized in Table 28 and Table 29, respectively. Under all shift magnitudes on the **360_real** dataset, all p-values are below the significance level ($\alpha = 0.01$). This confirms that both additional fine-tuning strategies successfully instill invariance to horizontal circular shifts, enabling the models to robustly preserve semantic alignment across shifted versions.

Table 28: **[Both Encoders, LoRA Fine-Tuning, OpenCLIP, LAION-400M]**, the Wilcoxon Signed-Rank test results under horizontal circular shift of various $\delta_j$ pixels for different CLIP models on the **360_real** dataset, where the null hypothesis is that $|s - s^{\delta_j}|$ is greater than or equal to the stability bound $\beta$, and the significance level ($\alpha$) is 0.01. The p-values less than $\alpha$ are in bold.

| $\lambda$ | ViT | $\beta$ | $\delta_j$ | $W/8$ | $2W/8$ | $3W/8$ | $4W/8$ | $5W/8$ | $6W/8$ | $7W/8$ |
|---|---|---|---|---|---|---|---|---|---|---|
| 0.9914 | B/32 | 1.7919 | statistic | 0 | 0 | 0 | 0 | 0 | 0 | 0 |
| | | | p-value | **0** | **0** | **0** | **0** | **0** | **0** | **0** |
| 0.9905 | B/16 | 1.6547 | statistic | 0 | 0 | 0 | 0 | 0 | 0 | 0 |
| | | | p-value | **0** | **0** | **0** | **0** | **0** | **0** | **0** |
| 0.9897 | L/14 | 1.4245 | statistic | 0 | 0 | 0 | 0 | 0 | 0 | 0 |
| | | | p-value | **0** | **0** | **0** | **0** | **0** | **0** | **0** |

Table 29: **[Image Encoder, Full Fine-Tuning, OpenCLIP, LAION-400M]**, the Wilcoxon Signed-Rank test results under horizontal circular shift of various $\delta_j$ pixels for different CLIP models on the **360_real** dataset, where the null hypothesis is that $|s - s^{\delta_j}|$ is greater than or equal to the stability bound $\beta$, and the significance level ($\alpha$) is 0.01. The p-values less than $\alpha$ are in bold.

| $\lambda$ | ViT | $\beta$ | $\delta_j$ | $W/8$ | $2W/8$ | $3W/8$ | $4W/8$ | $5W/8$ | $6W/8$ | $7W/8$ |
|---|---|---|---|---|---|---|---|---|---|---|
| 0.9995 | B/32 | 1.7919 | statistic | 0 | 0 | 0 | 0 | 0 | 0 | 0 |
| | | | p-value | **0** | **0** | **0** | **0** | **0** | **0** | **0** |
| 0.9995 | B/16 | 1.6547 | statistic | 0 | 0 | 0 | 0 | 0 | 0 | 0 |
| | | | p-value | **0** | **0** | **0** | **0** | **0** | **0** | **0** |
| 0.9995 | L/14 | 1.4245 | statistic | 0 | 0 | 0 | 0 | 0 | 0 | 0 |
| | | | p-value | **0** | **0** | **0** | **0** | **0** | **0** | **0** |

To further compare the semantic evaluation capability of frozen and fine-tuned models, we computed CLIP scores ($s_i$) for all original 360-degree panoramic images in **360_real** using the frozen CLIP models and each of their fine-tuned variants. Boxplots in Fig. 13 and Fig. 14 show the distribution of original image scores under four settings: frozen CLIP, LoRA (image encoder only), LoRA (image and text encoders), and full fine-tuning (image encoder).

Across all architectures, LoRA applied to both the image and text encoders preserves more of the frozen CLIP model's original semantic evaluation behavior than fine-tuning the image encoder alone. Full fine-tuning of the image encoder exhibits semantic evaluation performance similar to LoRA applied to the image encoder only, but requires substantially more trainable parameters. LoRA on the image encoder only remains the most parameter-efficient method, offering a good balance between preserved original semantic evaluation performance and training cost. A detailed comparison of GPU memory and trainable parameter counts for all methods is provided in Table 30.

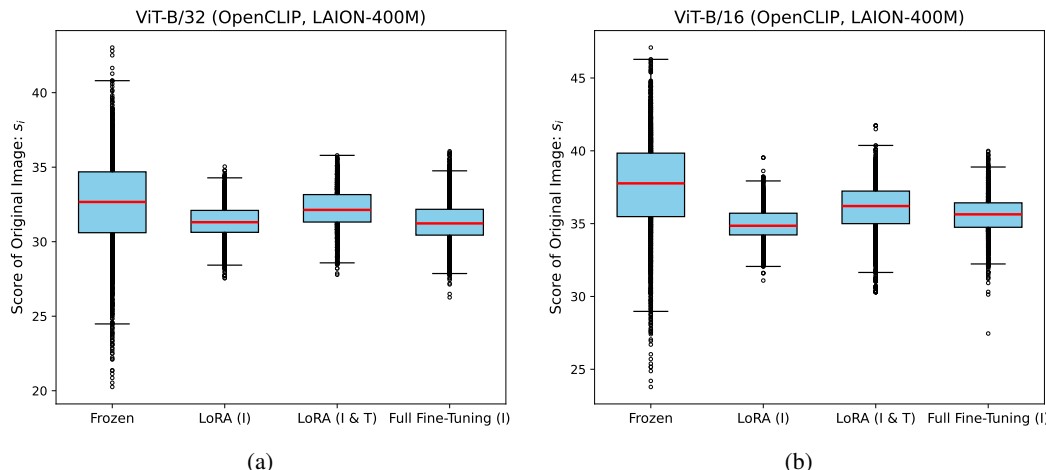

Figure 13: CLIP scores of original 360-degree panoramic images using a frozen CLIP model and its three fine-tuned versions with different fine-tuning methods. (I) and (I & T) denote fine-tuning on image encoder and both encoders, respectively.

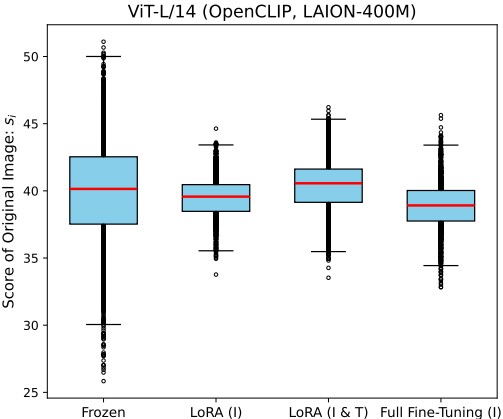

Figure 14: CLIP scores of original 360-degree panoramic images using a frozen CLIP model and its three fine-tuned versions with different fine-tuning methods. (I) and (I & T) denote fine-tuning on image encoder and both encoders, respectively.

Table 30: Comparison of different fine-tuning methods under the same training setting, where GPU memory is calculated when batch size is set to 16. The best results are highlighted.

| ViT | Methods | LoRA (Image) | LoRA (Image and Text) | Full Fine-Tuning (Image) |
|---|---|---|---|---|
| B/32 | GPU Memory (GB) | **1.9** | 2.7 | 3.3 |
| | Number of Trainable Parameters (M) | **0.52** | 0.86 | 87.85 |
| B/16 | GPU Memory (GB) | **3.8** | 4.5 | 6.1 |
| | Number of Trainable Parameters (M) | **0.52** | 0.86 | 86.19 |
| L/14 | GPU Memory (GB) | **11.3** | 12.2 | 18.5 |
| | Number of Trainable Parameters (M) | **1.38** | 1.89 | 303.97 |

## J    WHY SIGLIP IS NOT SUITABLE AS A SEMANTIC EVALUATOR?

Although SigLIP (Zhai et al., 2023) replaces the contrastive loss used by CLIP with a simpler and more scalable sigmoid loss, the SigLIP score is not currently used as a metric for evaluating image-text semantic alignment. In contrast, CLIP models have become the standard evaluators in this setting. Notably, CLIPScore (Hessel et al., 2021) demonstrates that CLIP-based similarity scores achieve the highest correlation with human judgments in image captioning evaluation. To investigate whether SigLIP can serve as a semantic evaluator, we conducted a comparison experiment between SigLIP (ViT-L-16, trained on WebLI (Chen et al., 2022a)) and CLIP (ViT-L-14, trained on LAION-400M).

We first constructed a dataset consisting of 500 text prompts generated by ChatGPT (OpenAI, 2025) and their corresponding perspective images ($1024 \times 512$ resolution) synthesized with SDXL (Podell et al., 2023), which we refer to as *per_syn*. Fig. 15 shows examples of the image-text pairs together with horizontally flipped and circular-shifted versions. Unlike 360-degree panoramic images, circular shifts in perspective images introduce clear semantic distortions, providing a controlled way to test whether a model's score meaningfully reflects semantic alignment.

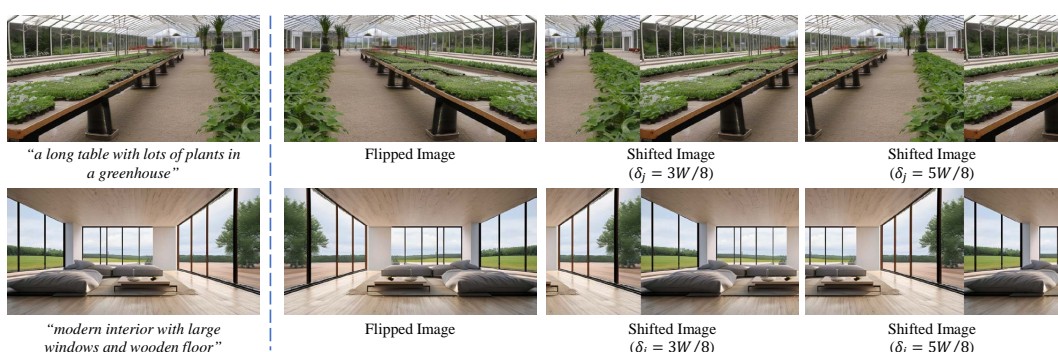

Figure 15: Examples of perspective image-text pairs in *per_syn*, and the horizontally flipped and circular-shifted versions of corresponding images. $\delta_j$ and $W$ denote the shift distance and the image width, respectively.

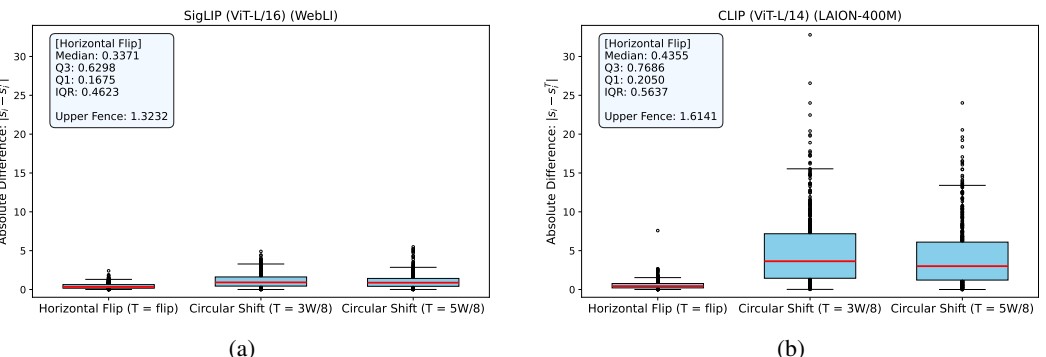

Figure 16: Boxplots of absolute score differences ($|s_i - s_i^T|$) under three diverse transformations for (a) SigLIP and (b) CLIP models on the *per_syn* dataset.

Following procedure in Sec. 2.2, we present the distribution of absolute score differences ($|s_i - s_i^{flip}|$) for the SigLIP and CLIP models on the leftmost side of each subfigure in Fig. 16. The stability bounds $\beta$ (defined as the upper fence in Sec. 2.2) for SigLIP and CLIP are also reported in the corresponding text boxes. Using these bounds, we then evaluated how each model responds to two circular-shifted magnitudes by conducting one-sided Wilcoxon Signed-Rank tests on $|s - s^{\delta_j}|$. The results are summarized in Table 31.

A reliable semantic evaluator should assign substantially different scores to circular-shifted images, which exhibit severe semantic distortions. This behavior is clearly observed in CLIP: all p-values

Table 31: **[SigLIP (ViT-L-16) VS. CLIP (ViT-L-14)]**, the Wilcoxon Signed-Rank test (Wilcoxon, 1945) results under horizontal circular shift of various $\delta_j$ pixels for SigLIP and CLIP models on the *per_syn* dataset, where the null hypothesis ($H_0$) is that $|s - s^{\delta_j}|$ is greater than or equal to the stability bound $\beta$, and the significance level ($\alpha$) is 0.01. The p-values less than $\alpha$ are in bold.

| ViT | $\beta$ | $\delta_j$ | $3W/8$ | $5W/8$ |
|---|---|---|---|---|
| SigLIP (L/16) | 1.3232 | *statistic* | 41288 | 33681 |
| | | *p-value* | **0** | **0** |
| CLIP (L/14) | 1.6141 | *statistic* | 108017 | 103635 |
| | | *p-value* | 1 | 1 |

exceed the significance level ($\alpha = 0.01$), indicating that CLIP effectively detects semantic mismatch. SigLIP, however, does not exhibit this expected behavior. Across both shift magnitudes, its p-values remain below $\alpha$ (see Table 31), meaning SigLIP assigns similar scores to the original and semantically corrupted images. The boxplots in Fig. 16 further reveal that, unlike CLIP, SigLIP's score differences under circular shifts do not significantly widen relative to those under horizontal flip.

These findings demonstrate that SigLIP's scoring mechanism does not reliably reflect image-text semantic alignment, even under strong semantic distortions. Consequently, SigLIP is not suitable as a quantitative semantic evaluator for our statistical probing framework, which requires a continuous, stable, and semantically meaningful similarity metric.

# K MORE QUANTITATIVE RESULTS

## K.1 360-DEGREE TEXTUAL SEMANTICS

To quantitatively demonstrate that CLIP models effectively leverage explicit 360-degree textual identifiers, we computed the average values of the original score and the generic score across the two paired image-text datasets (*360_real* and *360_syn*). The results, reported in Table 32 and Table 33, show that for all CLIP configurations, the average original score is substantially higher than the corresponding generic score. This confirms that CLIP models rely strongly on explicit 360-degree format cues in text, providing clear quantitative evidence of their understanding of 360-degree textual semantics.

Table 32: **[OpenCLIP, LAION-400M]** [$V^* =$"*<360panorama>,* "], the average values ($\bar{s}$ and $\bar{s}^u$) of original score $s$ and generic score $s^u$ on the two paired image-text datasets (*360_real* and *360_syn*).

| ViT | 360_real | | | 360_syn | | |
|---|---|---|---|---|---|---|
| | $V^*$ | $U^* =$"" | $U^* =$"*image,* " | $V^*$ | $U^* =$"" | $U^* =$"*image,* " |
| | $\bar{s}$ | $\bar{s}^u$ | $\bar{s}^u$ | $\bar{s}$ | $\bar{s}^u$ | $\bar{s}^u$ |
| B/32 | 32.5204 | 23.4912 | 24.2359 | 28.9801 | 22.4678 | 23.1340 |
| B/16 | 37.5371 | 27.1522 | 28.6953 | 31.1695 | 27.4186 | 29.1830 |
| L/14 | 39.8776 | 26.0096 | 26.9339 | 37.4755 | 26.3478 | 27.0026 |

Table 33: **[OpenCLIP, LAION-400M]** [$V^* =$"*<360panorama>,* "], the average values ($\bar{s}$ and $\bar{s}^u$) of original score $s$ and generic score $s^u$ on the two paired image-text datasets (*360_real* and *360_syn*).

| ViT | 360_real | | | 360_syn | | |
|---|---|---|---|---|---|---|
| | $V^*$ | $U^* =$"*photo,* " | $U^* =$"*picture,* " | $V^*$ | $U^* =$"*photo,* " | $U^* =$"*picture,* " |
| | $\bar{s}$ | $\bar{s}^u$ | $\bar{s}^u$ | $\bar{s}$ | $\bar{s}^u$ | $\bar{s}^u$ |
| B/32 | 32.5204 | 23.7829 | 23.8902 | 28.9801 | 22.8859 | 22.6689 |
| B/16 | 37.5371 | 27.4826 | 27.7713 | 31.1695 | 27.6305 | 27.7945 |
| L/14 | 39.8776 | 26.4968 | 26.5376 | 37.4755 | 26.2792 | 26.6744 |

## K.2 360-DEGREE VISUAL SEMANTICS

To provide more direct quantitative evidence regarding the models' understanding of 360-degree visual semantics, we present CLIP scores of original 360-degree panoramic images and their corresponding horizontally circular-shifted versions using frozen and fine-tuned CLIP models in Fig. 17, Fig. 18, and Fig. 19.

For frozen CLIP models, the scores vary noticeably across different circular shifts, indicating that they fail to preserve stable semantic alignment under this transformation, consistent with our statistical results. In contrast, our fine-tuned models remain stable scores across all shift magnitudes, demonstrating a stable and robust understanding of 360-degree visual semantics.

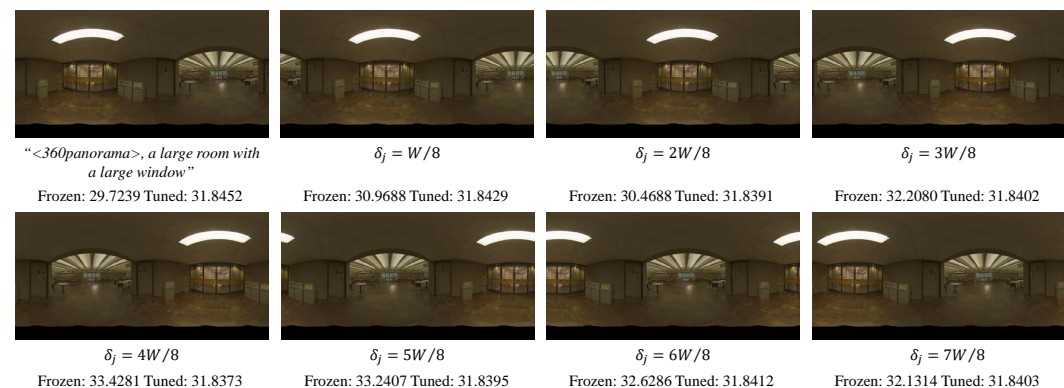

Figure 17: **[ViT-B/32, OpenCLIP, LAION-400M]**, CLIP scores of an original 360-degree panoramic image and its corresponding horizontally circular-shifted versions using frozen and fine-tuned CLIP models, respectively. $\delta_j$ and $W$ denote the shift distance and the image width, respectively.

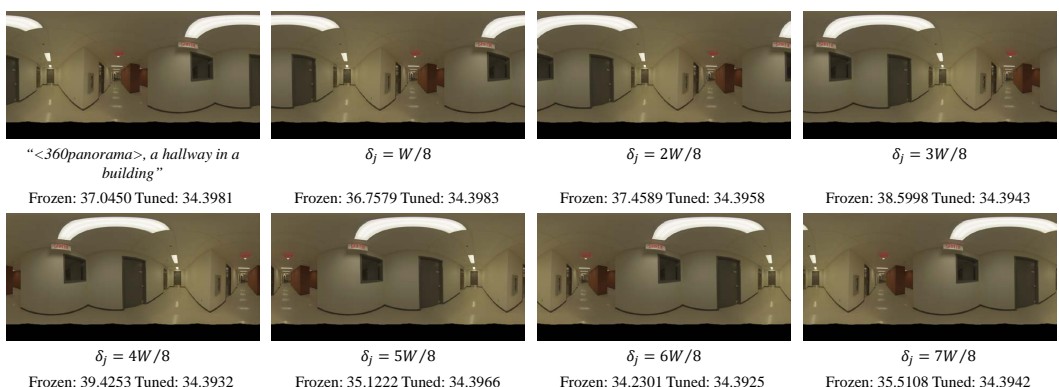

Figure 18: **[ViT-B/16, OpenCLIP, LAION-400M]**, CLIP scores of an original 360-degree panoramic image and its corresponding horizontally circular-shifted versions using frozen and fine-tuned CLIP models, respectively. $\delta_j$ and $W$ denote the shift distance and the image width, respectively.

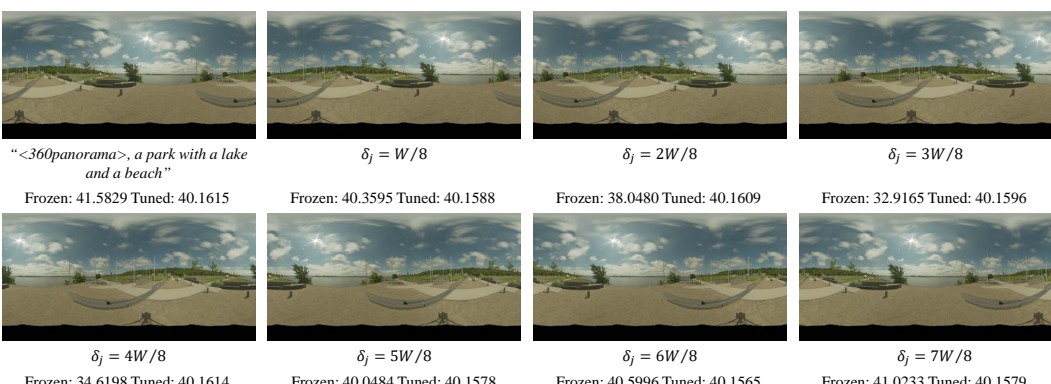

Figure 19: **[ViT-L/14, OpenCLIP, LAION-400M]**, CLIP scores of an original 360-degree panoramic image and its corresponding horizontally circular-shifted versions using frozen and fine-tuned CLIP models, respectively. $\delta_j$ and $W$ denote the shift distance and the image width, respectively.

