# OpenReview forum: "Probing CLIP's Comprehension of 360-Degree Textual and Visual Semantics"
_ICLR.cc/2026/Conference — Submitted to ICLR 2026_

### Official Review · Reviewer_3QfH · 2025-10-27

**Soundness:** 2
**Presentation:** 2
**Contribution:** 2
**Rating:** 4
**Confidence:** 4

**Summary:**

The paper's title is "probing CLIP's comprehension of 350-degree textual and visual semantics". The paper presents several themes, including 360-dgeree panoramic image generation (040), accurate evaluation of semantic alignment between generated 360-degree panoramic images (Line 050), and CLIP models' comprehension ability in 360-degree panoramic image-text pairs (Line 072). These themes make the paper less focused. Nevertheless, the paper studies CLIP models in term of their ability to understand panoramic images. It adopts various statistical tools to investigate this matter. It horizontally shifts panoramic images and measures how consistent CLIP models output scores. It concludes that CLIP models can understand keywords such as "a 360 degree view of" and "panorama", but does not process robustness in producing consistent scores on horizontally shifted panoramic images. It adopts LoRA to adapt the visual encoder, enhancing this robustness.

**Strengths:**

Below are notable strengths of this paper.

- The variety of statistical tools are interesting.
- Technical design choices are generally well thought-out, e.g., the null hypotheses, dataset construction, etc.
- Using LoRA to empower CLIP models' robustness to horizontal-shift panorama is interesting.

**Weaknesses:**

Below are notable weaknesses of this paper.

- The focus of this paper is not clear. Line 011 and the first paragraph of Introduction leave an impression that this paper focuses on generating 360-degree panoramic images. However, Line 049 motivates evaluating semantic alignments between generated 360-degree panoramic images, leaving an impression that the work focuses on developing methods to evaluate generative models' performance in terms of semantic alignments of their generated images. Quite confusingly, Line 051 and Line 072 state that the the work particularly focuses on CLIP models and studies their understanding ability of panoramic images.

- As the paper is motivated for accurate evaluation of semantic alignment between generated 360-degree panoramic images, it is not convincing to only focus on CLIP (Line 073). Given the existing of powerful Multimodal Large Language Models (MLLMs), it is natural to ask whether these MLLMs can serve as the evaluator.

- Line 064 states "raises questions about their applicability to evaluating 360-degree panoramic image-text pairs, which present fundamentally different characteristics". Did the authors justify that 360-degree panoramic image-text pairs "present fundamentally different characteristics" from CLIP's pretraining data? Do the CLIP's pretraining data contain panoramic images? This is an important point, as the analysis of CLIP's understanding ability on panoramic images can be greatly determined by whether the CLIP's pretraining dataset contains panoramic images and the amounts / portion of such images

- Line 035 and 068 mention "360x180". What does 180 mean?

- In Equation (1), it is unclear why multiplying the constant 100. It is unclear either why using a max operation. The paper does not explain these

- Table 1 presents some statistical testing results but does not explain what data are used in the test. Specifically, it states that it uses "two paired image-text datasets (360_real and 360_syn) but does not explain how they are constructed and what data they contain. (Okay, the paper talks about them in later text, Section 4.1. But the current presentation causes confusions, i.e., using the acronyms without defining them.)

- It is questionable whether the current design of beta (Line 240) is a good choice. It is based on the difference of CLIP scores between images and their left-right flips. To align with horizontal shifts in the study of 360-degree visual semantic, isn't it a better choice to properly horizontally shift images to derive beta?

- The paper uses LoRA to adapt CLIP models for gaining a robust understanding of understanding of 360-degree visual semantics. But the paper does not explore how it helps evaluate generated panoramic images, which is the theme of the paper (Line 049).

**Questions:**

The reviewer asks the authors to address each point in weaknesses listed above and does not repeat them in this Questions box.

---

> ### Author Response · Authors · 2025-11-21
> **Response to Reviewer 3QfH**
>
> **W1 & W8:**
> We would like to clarify the central theme of our work, which we believe is single and coherent. As stated in the title, our focus is on **Probing CLIP's Comprehension of 360-Degree Textual and Visual Semantics.** We are confident in the paper's clarity, a view shared by other reviewers. For instance, Reviewer W71t comments that, *All these presentation are clear in the draft and writing is well-organized.*
>
> The discussion of 360-degree panorama generation serves as motivation, since existing methods [3-7] universally rely on CLIP for semantic evaluation. This context leads directly to our central research question: **To what extent can standard CLIP models, predominantly trained on perspective image–text pairs, comprehend the distinct semantics inherent in 360-degree panoramic image–text pairs?**
> Our findings show that, while CLIP models possess a strong grasp of 360-degree textual semantics, they fail to maintain stable alignment scores under horizontal circular shifts. This instability reveals a critical limitation in CLIP's understanding of 360-degree visual semantics, which directly motivates the fine-tuning component of our work.
> Therefore, the purpose of our LoRA-based fine-tuning framework is to directly address this specific failure mode: the lack of robustness to circular shifts. As shown in Sec. K.2 of the revised paper, the fine-tuned models produce stable CLIP scores across all shift magnitudes, while the frozen CLIP models do not. This experiment validates our diagnosis and demonstrates that the identified limitation is correctable. We also emphasize that this enhancement comes with a trade-off, namely, a slight degradation in the original semantic evaluation performance.
>
> **W2:**
> CLIP is the *de facto* standard quantitative evaluator used by current text-driven 360-degree panorama generation pipelines (e.g., [3-7]). Understanding its limitations therefore directly benefits existing practice. MLLMs, by contrast, typically provide qualitative judgments rather than a stable similarity metric comparable to CLIP’s cosine similarity, and they are not yet used in 360-degree panoramic evaluation benchmarks. We fully agree that exploring MLLMs as evaluators for 360-degree content is an intriguing research direction. To acknowledge this, we have added this point to the future work section (see Sec. 6).
>
> **W3:**
> The CLIP models evaluated in our paper are pre-trained on large-scale web-scraped datasets, including LAION-400M, LAION-2B, and OpenAI's WIT. By nature, these datasets consist predominantly of standard 2D perspective image-text pairs, reflecting the natural distribution of images on the Internet. As a result, 360-degree panoramic image-text pairs represent a statistically negligible minority in the training data. This data imbalance directly motivates our statement that "the predominant training of CLIP models on perspective image-text pairs raises questions about their applicability to evaluating 360-degree panoramic image-text pairs, which present fundamentally different characteristics".
>
> In addition, we emphasize that our experiments include CLIP models (ViT-B/32, ViT-B/16, and ViT-L/14) pre-trained on LAION-400M (see Sec. 4) and LAION-2B (see Sec. F), respectively. For both pre-training datasets, all CLIP models effectively leverage explicit textual identifiers, indicating an understanding of 360-degree textual semantics. However, they fail to robustly maintain semantic alignment under horizontal circular shifts, demonstrating limited comprehension of 360-degree visual semantics. These results further show that simply increasing the scale of pre-training data (from 400M to 2B samples) does not improve robustness to 360-degree visual semantics.
>
> **W4:**
> The term "360° × 180°" denotes a full spherical field of view, covering 360 degrees horizontally and 180 degrees vertically.
>
> **W5:**
> These follow the standard implementation of the CLIP score metric. Please refer to this online documentation (https://lightning.ai/docs/torchmetrics/stable/multimodal/clip_score.html) and the original CLIPScore paper [9] for details.
>
> **W6:**
> To avoid confusion, we have added a reference in the caption of Table 1.
>
> **W7:**
> To explain our motivation intuitively, consider a simple analogy: when determining a diagnostic threshold for a medical test, we would not estimate this threshold using measurements from the entire population, because it contains both healthy and ill individuals. Instead, the threshold is determined from healthy individuals only, so that it reflects the natural baseline variation.
>
> Similarly, we use horizontal flips (a semantic-preserving transformation for CLIP, as shown in [11]) to estimate a baseline bound $\beta$ that reflects CLIP's normal score variation under a transformation that should not change semantics. This bound then serves as a reference to evaluate whether CLIP scores remain stable under horizontal circular shifts in 360-degree panoramas.

---

### Official Review · Reviewer_3aJi · 2025-10-27

**Soundness:** 3
**Presentation:** 2
**Contribution:** 2
**Rating:** 4
**Confidence:** 3

**Summary:**

This paper systematically investigates the semantic understanding capability of the pre-trained CLIP model regarding 360° panoramic images and related textual descriptions, particularly its limitations in circular/360-degree visual semantics. The authors find that horizontal circular shifts of panoramic images cause significant fluctuations in the CLIP score, indicating the model's lack of intrinsic understanding of the geometric property of such images. To address this issue, the authors propose a lightweight fine-tuning framework based on Low-Rank Adaptation (LoRA), designed to enhance the CLIP image encoder's perception of 360° visual semantics.

**Strengths:**

1. The problem is well-defined and the motivation is clear: The research addresses a practical limitation of CLIP in panoramic vision tasks, which is highly relevant. Through a series of methods and designs, the authors measure the model's sensitivity to both semantics (e.g., 360°-related cues in prompts) and visual features in panoramic images. The conclusion—that CLIP is more sensitive to semantic information but less capable in capturing and extracting panoramic vision features—is reasonable and logically sound.

2. The method design is clever: Using image augmentation and incorporating constraints effectively enhances CLIP's feature extraction capability for 360° panoramic images and improves semantic-visual alignment.

**Weaknesses:**

The main weakness of this work is the lack of demonstrated applicability in downstream scenarios and insufficient experimental results, which primarily rely on inspection. Firstly, providing more quantitative results, rather than the binary 0/1 outcomes from the inspection, would be more persuasive. Secondly, it would be significantly better to show positive results on downstream tasks, such as 360° image retrieval/generation or visual question answering. If show promising results, I will consider raise my score.

**Questions:**

1. The current experiments focus on indoor and cityscape panoramas (e.g., Laval). It is recommended to test the model's generalization on more diverse scenes, such as natural landscapes or dynamic environments.

2. The current method only fine-tunes the image encoder. Since 360° semantics also involve textual understanding, future work could explore jointly fine-tuning the text encoder or designing more fine-grained text prompts.

---

> ### Author Response · Authors · 2025-11-21
> **Response to Reviewer 3aJi**
>
> **W1:**
> Thank you for this helpful suggestion. In the revised paper, we have added additional quantitative results in Sec. K. Specifically, we now report the average CLIP scores for the original prompts and their generic counterparts on the paired image-text datasets (*360\_real* and *360\_syn*). These results clearly show that all CLIP models assign significantly higher scores to prompts containing explicit 360-degree identifiers, providing strong quantitative evidence that they effectively leverage 360-degree textual semantics.
>
> Furthermore, we have also reported the CLIP scores of original 360-degree panoramic images and their corresponding horizontally circular-shifted versions for both frozen and fine-tuned CLIP models. These quantitative results demonstrate that frozen CLIP models fail to maintain stable semantic alignment under circular shifts, while our fine-tuned models exhibit substantially improved robustness and a clear understanding of 360-degree visual semantics.
>
> **W2:**
> Thank you for this insightful suggestion. Our work centers on the following question: **To what extent can standard CLIP models, predominantly trained on perspective image-text pairs, comprehend the distinct semantics inherent in 360-degree panoramic image-text pairs?** Our findings show that, while CLIP models already possess a strong grasp of 360-degree textual semantics, they lack robust comprehension of 360-degree visual semantics, which motivates our fine-tuning framework.
>
> Our fine-tuning method is specifically designed to enhance visual semantic invariance under horizontal circular shifts. As shown in Sec. K.2 of the revised paper, the fine-tuned models produce stable CLIP scores across all shift magnitudes, while the frozen CLIP models do not. This demonstrates a clear and positive improvement in 360-degree visual semantic understanding. However, enhancing robustness to circular shifts necessarily alters the feature distribution and semantic evaluation behavior of the original CLIP model. Many downstream tasks, such as image-text retrieval or visual question answering, rely on these original CLIP features, and applying our fine-tuned models directly may therefore degrade their performance. For these reasons, we view downstream task evaluation as an important but orthogonal direction, and we leave the exploration of this aspect to further work.
>
> **Q1:**
> Thank you for this helpful suggestion. To assess generalization beyond indoor and cityscape panoramas, we have utilized Diffusion360 to generate 500 360-degree panoramas of natural landscapes, with text prompts produced by ChatGPT. We refer to this new dataset as *360\_nature*.
>
> Following the procedure in Sec. 2.2, the stability bounds $\beta$ on the *360\_nature* dataset for ViT-B/32, ViT-B/16, and ViT-L/14 are 1.7401, 1.9949, and 1.7896, respectively. Using these stability bounds, we have conducted one-sided Wilcoxon Signed-Rank test for CLIP models trained on LAION-400M. As reported in Table 25 (Sec. H.1 of the revised paper), all p-values were consistently below the significance level ($\alpha$ = 0.01), indicating that the fine-tuned CLIP models continue to exhibit a robust understanding of 360-degree visual semantics on natural-landscape scenes. These results demonstrate the strong generalization capability of our fine-tuned models to more diverse scene types.
>
> **Q2:**
> Thank you for this valuable suggestion. In the revised paper, we have added a new analysis in Sec. I examining the effect of applying LoRA to both the image and text encoders. In addition, we also explored full fine-tuning of the image encoder.
>
> Across all CLIP architectures, all fine-tuned models achieved p-values below $\alpha = 0.01$ in our Wilcoxon Signed-Rank tests, confirming that every fin-tuning strategy successfully instills invariance to horizontal circular shifts and enhances the model's understanding of 360-degree visual semantics. We have further compared the models' semantic evaluation performance on original 360-degree panoramic images, and analyzed GPU memory usage and parameter counts. Our findings show that the LoRA applied to both encoders best preserves the original forzen CLIP model's semantic evaluation performance. Full fine-tuning of the image encoder performs similarly to the LoRA on the image encoder only, but requires significantly more parameters. The LoRA applied only to the image encoder remains the most parameter-efficient method, providing a good trade-off between effectiveness and training cost.

---

### Official Review · Reviewer_W71t · 2025-11-01

**Soundness:** 4
**Presentation:** 3
**Contribution:** 2
**Rating:** 4
**Confidence:** 4

**Summary:**

This paper explores the CLIP models' comprehension abilities of 360-degree image's visual and textual semantics. To probe CLIP's such visual and textual semantics, the paper set up an evaluation method in two ways: keyword manipulation for textual cues, and horizontal circular shift of 360-degree images for visual cues. Based on these evaluations, first CLIP is able to exploit the textual cue such as "360-degree image" for a better alignment with the corresponding image. However, second, The invariance does not robustly hold with respect to the image's circular horizontal shift, indicating CLIP's limited comprehension of 360 degree visual semantics. To remedy this, authors propose to fine-tune the CLIP models by introducing LoRA adapters on visual encoders to introduce invariance to such horizontal shifts in the images. Experiments show the improve performance of understanding visual cues of 360-degree circular shifts across different CLIP models.

**Strengths:**

This paper investigates the CLIP's understanding capabilities of textual and visual semantics, which is underexplored in literature and derived a meaningful observations and solution. Specifically, the findings that (1) CLIP can exploit "360-degree image-specific" cues in the text prompt, rather than a generic prompt like "a photo of" for a better alignment to its image, and (2) CLIP's alignment especially lacks invariance to 360-degree image's circular horizontal shift would be valuable. All these presentation are clear in the draft and writing is well-organized.

**Weaknesses:**

While the motivation and observation presented in the paper are quite strong, but the major concern lies in the technical side. The devised solution to improve the CLIP's understanding of visual semantics appears too straightforward and leads to expected results; fine-tuning CLIP on shifted images can naturally enhances its robustness to such shifts during inference. In addition, although the method adopts fine-tuning with LoRA, it is important to include different fine-tuning methodologies as well, such as full fine-tuning of both encoders or each encoder individually, to measure the respective effectiveness. Further reasoning and analysis for the fine-tuning part are expected to strengthen its technical solidity of this work.

**Questions:**

Based on the weaknesses stated above, further analyses on the fine-tuning methodology and the fine-tuned models are expected. First, how the different fine-tuning strategies affect the image and text understanding capabilities in 360-degree images? Second, regarding robustness in visual semantics, can the fine-tuned models generalize to shift magnitudes unseen during training (e.g., when the horizontal circular shift applied at test time exc eeds the range observed during training)? In addition, one natural question is that does the scale of pre-training data of CLIP models affect the robustness of visual and textual semantics understanding after fine-tuning?

---

> ### Author Response · Authors · 2025-11-21
> **Response to Reviewer W71t**
>
> **Q1. First, how the different fine-tuning strategies affect the image and text understanding capabilities in 360-degree images?**
>
> Thank you for this good question. We have added a new analysis in Sec. I of the revised paper that compares multiple fine-tuning strategies and examines their impact on CLIP's ability to understand 360-degree visual semantics. In addition to the LoRA-based image-encoder tuning used in the main paper, we also evaluate (1) LoRA applied to both the image and text encoders, and (2) full fine-tuning of the image encoder.
>
> Across all CLIP architectures, all fine-tuned models achieved p-values below $\alpha = 0.01$ in our Wilcoxon Signed-Rank tests, confirming that every fin-tuning strategy successfully instills invariance to horizontal circular shifts and enhances the model's understanding of 360-degree visual semantics. We have further compared the models' semantic evaluation performance on original 360-degree panoramic images, and analyzed GPU memory usage and parameter counts. Our findings show that the LoRA applied to both encoders best preserves the original forzen CLIP model's semantic evaluation performance. Full fine-tuning of the image encoder performs similarly to the LoRA on the image encoder only, but requires significantly more parameters. The LoRA applied only to the image encoder remains the most parameter-efficient method, providing a good trade-off between effectiveness and training cost.
>
> **Q2. Second, regarding robustness in visual semantics, can the fine-tuned models generalize to shift magnitudes unseen during training (e.g., when the horizontal circular shift applied at test time exceeds the range observed during training)?**
>
> Thank you for this question. To evaluate generalization to unseen shift magnitudes, we have modified the training procedure so that the shift distance $\Delta$ was randomly selected from $\{0, 32, 64, \dots, 992 \}$. We have then carried out the Wilcoxon Signed-Rank test under horizontal circular shift of 110, 210, 310, 410, 510, 610, and 710 pixels on both the *360\_real* and *360\_syn* datasets. As shown in Sec. H.2 of the revised paper, all p-values at these seven unseen shift magnitudes were consistently below the significance level ($\alpha$ = 0.01), demonstrating strong generalization capability of the fine-tuned model.
>
> **Q3. In addition, one natural question is that does the scale of pre-training data of CLIP models affect the robustness of visual and textual semantics understanding after fine-tuning?**
>
> Thank you for this insightful question. We have evaluated CLIP models (ViT-B/32, ViT-B/16, and ViT-L/14) pre-trained on LAION-400M (see Sec. 4) and LAION-2B (see Sec. F), respectively. For both pre-training datasets, all CLIP models effectively leverage explicit textual identifiers, indicating an understanding of 360-degree textual semantics. However, they fail to robustly maintain semantic alignment under horizontal circular shifts, demonstrating limited comprehension of 360-degree visual semantics. These results show that increasing the scale of pre-training data alone (from 400M to 2B samples) does not improve robustness to 360-degree visual semantics.
>
> With our proposed LoRA-based fine-tuning, all CLIP models, whether pre-trained on LAION-400M or LAION-2B, exhibit robustness to horizontal circular shifts, reflecting enhanced comprehension of 360-degree visual semantics.

---

> ### Comment · Reviewer_W71t · 2025-11-27
>
> Dear authors,
> Thank you for providing additional findings on my questions.
>
> My concerns have been resolved, so I increased the score to 6.
>
> Below are some additional suggestions to further improve the paper:
>
> - As pointed out by Reviewer 3aJi, it would be more interesting to show the impact on the models on some 3D-related downstream tasks after training.
>
> - To improve the comprehensive of the work, different family of CLIP models, such as SigLIP or other kind of models can also be tested. In addition, it would be helpful to see whether MLLMs also share the same issue as in CLIP.

---

> > ### Author Response · Authors · 2025-12-02
> > **Response to Reviewer W71t**
> >
> > Thank you for increasing the score and for these valuable suggestions.
> >
> > (1) Regarding 3D-related downstream tasks: as noted in our response to Reviewer 3aJi, our work centers on the following question: **To what extent can standard CLIP models, predominantly trained on perspective image-text pairs, comprehend the distinct semantics inherent in 360-degree panoramic image-text pairs?** Our findings show that, while CLIP models already possess a strong grasp of 360-degree textual semantics, they lack robust comprehension of 360-degree visual semantics, which motivates our fine-tuning framework.
> >
> > Our fine-tuning method is specifically designed to enhance visual semantic invariance under horizontal circular shifts. As shown in Sec. K.2 of the revised paper, the fine-tuned models produce stable CLIP scores across all shift magnitudes, while the frozen CLIP models do not. This demonstrates a clear and positive improvement in 360-degree visual semantic understanding. However, enhancing robustness to circular shifts necessarily alters the feature distribution and semantic evaluation behavior of the original CLIP model. Many downstream tasks rely on these original CLIP features, and applying our fine-tuned models directly may therefore degrade their performance. For these reasons, we view downstream task evaluation as an important but orthogonal direction, and we leave the exploration of this aspect to future work.
> >
> > (2) We have added a comparison with SigLIP in Sec. J of the revised manuscript. Our findings demonstrate that SigLIP scores do not reliably reflect image-text semantic alignment, and therefore SigLIP cannot be used as an evaluator for our probing framework. Current multimodal LLMs typically produce qualitative textual responses rather than a stable quantitative similarity score comparable to CLIP's cosine similarity. Moreover, MLLMs are not yet integrated into standard evaluation protocols for this task, and the lack of established 360-degree panoramic benchmarks currently makes them less practical in this setting. We acknowledge that exploring MLLMs as evaluators for 360-degree content is an intriguing research direction, and we have added this point to the future work section (see Sec. 6).

---

### Official Review · Reviewer_UWJ9 · 2025-11-03

**Soundness:** 2
**Presentation:** 3
**Contribution:** 2
**Rating:** 4
**Confidence:** 4

**Summary:**

This paper probes whether CLIP understands 360° panoramic semantics.

It defines two new notions:  textual semantics (explicit cues like “360 photo”) and visual semantics (invariance under circular shifts) — and tests them via statistical analysis.

Results show CLIP models rely on textual cuse but fail to maintain shift invariance.

A LoRA-based fine-tuning improves robustness but slightly degrades baseline performance
.

**Strengths:**

- This work clearly defines 360-degree textual semantics and 360-degree visual semantics, addressing a **novel and underexplored problem**.

- The presentation is clear and concise, making it easy to grasp the main ideas quickly.

- The proposed LoRA-based fine-tuning effectively instills shift invariance for 360-degree panoramic scenes.

- The paper provides comprehensive experiments and analyses to support its claims.

**Weaknesses:**

1. The overall method is complete and well studied. But the main concern is that the explored 360-degree visual setting represents a **relatively narrow scenario** and can be viewed as a special case of standard 2D images.

2. The proposed LoRA-based tuning is a commonly used technique, and thus the methodological novelty appears limited.

3. The paper relies on the original CLIP model, which is **somewhat outdated**; incorporating comparisons with more recent models such as SigLIP-V2 or Qwen-VL would strengthen the analysis.

4. While the experimental results are solid, they primarily serve as confirmatory findings rather than revealing deeper insights or unexpected behaviors.

**Questions:**

1. Would the same findings hold for SigLIP or multimodal LLMs (e.g., CLIP-based vision towers in LLaVA or Kosmos-2)?

2. Any visualization or interpretability on why CLIP fails under shifts?

---

> ### Author Response · Authors · 2025-11-21
> **Response to Reviewer UWJ9**
>
> **W1:** We would like to clarify that the 360-degree panoramic image is a genuinely distinct domain, not merely a special case of standard 2D images, and this importance is widely recognized in recent literature such as [1-6], which highlights that 360-degree panoramic content forms a rapidly expanding research ecosystem.
>
> **W2:** While LoRA itself is a commonly used adaptation technique, our contribution lies in the specific fine-tuning objective tailored to the 360-degree setting rather than in the adapter mechanism. Our method explicitly enforces invariance to horizontal circular shifts while regularizing the model to preserve its original semantic behavior, an objective motivated by the unique visual semantics of 360-degree panoramas. This formulation leads to new empirical insights, including a previously unreported trade-off between enhanced shift invariance and baseline semantic evaluation performance, which we believe constitutes meaningful methodological novelty well beyond the use of LoRA.
>
> **W3:** Our decision to focus on CLIP is motivated by its role as the *de facto* standard for semantic evaluation in current text-driven 360-degree panorama generation pipelines, as used in works such as [3-7]. Since CLIP is the tool the community currently depends on for quantitative alignment scores, understanding its behavior and improving its robustness directly benefit existing methodologies and ongoing research. Following your suggestion, we have added a comparison with SigLIP in Sec. J of the revised manuscript.
>
> **W4:** Beyond confirming expected trends, our experiments reveal several non-trivial and previously unreported behaviors. Specifically, we show that CLIP models effectively leverage explicit textual identifiers, highlighting the importance of including format-specific cues in prompts for accurate semantic evaluation of 360-degree panoramas. In addition, we demonstrate that CLIP models fail to robustly preserve semantic alignment under horizontal circular shifts. Furthermore, our fine-tuning analysis uncovers a trade-off between enforcing shift invariance and preserving baseline semantic evaluation performance. These findings provide deeper insights into CLIP's limitations on 360-degree panoramic content rather than merely confirming expected outcomes.
>
> **Q1:** Thank you for this insightful question. We address SigLIP and multimodal LLMs separately below.
>
> Although SigLIP [8] replaces CLIP's contrastive loss with a sigmoid loss, SigLIP scores are not currently used as quantitative metrics for image-text semantic alignment. In contrast, CLIP-based similarity is widely adopted, and CLIPScore [9] shows that CLIP similarities achieve the highest correlation with human judgments. To examine whether SigLIP could serve as an evaluator in our setting, we have conducted a controlled comparison between SigLIP (ViT-L/16, trained on WebLI [10]) and CLIP (ViT-L/14, LAION-400M). As detailed in Sec. J, SigLIP fails to distinguish original images from semantically distorted (horizontally circular-shifted) versions on our **per\_syn** dataset: all Wilcoxon p-values fall below $\alpha=0.01$, indicating that SigLIP assigns similar scores to both. In contrast, CLIP correctly identifies these semantic inconsistencies, with all p-values exceeding $\alpha$. These findings demonstrate that SigLIP scores do not reliably reflect image-text semantic alignment and therefore cannot used as evaluators for our probing framework.
>
> Current multimodal LLMs typically produce qualitative textual responses rather than a stable quantitative similarity score comparable to CLIP's cosine similarity. Moreover, MLLMs are not yet integrated into standard evaluation protocols for this task, and the lack of established 360-degree panoramic benchmarks currently makes them less practical in this setting. We acknowledge that exploring MLLMs as evaluators for 360-degree content is an intriguing research direction, and we have added this point to the future work section (see Sec. 6).
>
> **Q2:**
> Thank you for this question. Intuitively, CLIP fails under horizontal circular shifts because its original contrastive training objective does not impose any constraint that forces the CLIP score of an image-text pair to remain consistent across shifted variants. As a result, shifted variants of the same 360-degree panorama can produce unstable similarity scores.

---

> > ### Comment · Reviewer_UWJ9 · 2025-11-28
> >
> > I read the given paper list and acknowledge it is a valuable direction, please include these works also in the revised version.
> >
> > My concern regarding the comparison over Siglipv2 or other modern architectures is partly addressed, but I suggest the author include this comparison in the main body of paper with simple discussion in future version. Since the CLIP gradually not used for recent mllm works, the above concern is widely existing.
> > I preserve the opinion that current research should focus on more recent models. The quantitative similarity score of CLIP performs weak on scenes like text-rich scenes in image generation data. At current stage,  360-degree visual setting does not have simple altentive solution and I encourage the author thinking more about this topic.
> > For this current revised paper,  I tend to give boardline accept.

---

> > > ### Author Response · Authors · 2025-12-02
> > > **Response to Reviewer UWJ9**
> > >
> > > Thank you for these further comments. We will incorporate these works into the revised version. Following your suggestion, we will move the SigLIP comparison into the main paper with a brief discussion. We agree that extending the analysis to more recent models is an important direction and will continue to pursue this in future work. We also appreciate the additional insightful point regarding CLIP's limitations, which is valuable for guiding our future research.
> > >
> > > We appreciate your constructive feedback and your assessment of the revised manuscript as borderline acceptable.

---

### Author Response · Authors · 2025-11-21
**Response to all Reviewers**

We would like to thank all reviewers for their valuable feedback and constructive suggestions. We are grateful that multiple reviewers highlighted the novelty and importance of the problem we address, describing it as **novel** (Reviewer UWJ9), **underexplored** (Reviewer W71t), and **well-defined** (Reviewer 3aJi). We also appreciate the positive comments regarding our technical design choices and methodologies (Reviewer 3QfH).

We have carefully revised the manuscript in response to all comments, and provide detailed point-by-point replies below. For convenience, all references cited in the responses are listed here.

**References**

[1] Wang, C., Li, X., Qi, L., Lin, X., Bai, J., Zhou, Q. and Tong, Y., 2025. Conditional Panoramic Image Generation via Masked Autoregressive Modeling. arXiv preprint arXiv:2505.16862.

[2] Park, Minho, et al. "SphereDiff: Tuning-free Omnidirectional Panoramic Image and Video Generation via Spherical Latent Representation." arXiv preprint arXiv:2504.14396 (2025).

[3] Ni, J., Zhang, C.B., Zhang, Q. and Zhang, J., 2025. What Makes for Text to 360-degree Panorama Generation with Stable Diffusion?. In Proceedings of the IEEE/CVF International Conference on Computer Vision (pp. 16555-16564).

[4] Yang, L., Duan, H., Zhu, Y., Liu, X., Liu, L., Xu, Z., Ma, G., Min, X., Zhai, G. and Le Callet, P., 2025, October. Omni2: Unifying omnidirectional image generation and editing in an omni model. In Proceedings of the 33rd ACM International Conference on Multimedia (pp. 10103-10112).

[5] H. Wang, X. Xiang, W. Xia and J.-H. Xue, "A Survey on Text-Driven 360-Degree Panorama Generation," in IEEE Transactions on Circuits and Systems for Video Technology, doi: 10.1109/TCSVT.2025.3628738.

[6] Sun, X., Xu, M., Li, S., Ma, S., Deng, X., Jiang, L. and Shen, G., 2025. Spherical Manifold Guided Diffusion Model for Panoramic Image Generation. In Proceedings of the Computer Vision and Pattern Recognition Conference (pp. 5824-5834).

[7] Zhang, C., Wu, Q., Gambardella, C.C., Huang, X., Phung, D., Ouyang, W. and Cai, J., 2024. Taming stable diffusion for text to 360 panorama image generation. In Proceedings of the IEEE/CVF Conference on Computer Vision and Pattern Recognition (pp. 6347-6357).

[8] Zhai, X., Mustafa, B., Kolesnikov, A. and Beyer, L., 2023. Sigmoid loss for language image pre-training. In Proceedings of the IEEE/CVF international conference on computer vision (pp. 11975-11986).

[9] Hessel, J., Holtzman, A., Forbes, M., Le Bras, R. and Choi, Y., 2021, November. Clipscore: A reference-free evaluation metric for image captioning. In Proceedings of the 2021 conference on empirical methods in natural language processing (pp. 7514-7528).

[10] Chen, X., Wang, X., Changpinyo, S., Piergiovanni, A.J., Padlewski, P., Salz, D., Goodman, S., Grycner, A., Mustafa, B., Beyer, L. and Kolesnikov, A., 2022. Pali: A jointly-scaled multilingual language-image model. arXiv preprint arXiv:2209.06794.

[11] Wang, Tiancheng, Yuguang Yang, Linlin Yang, Shaohui Lin, Juan Zhang, Guodong Guo, and Baochang Zhang. "CLIP in Mirror: Disentangling text from visual images through reflection." Advances in Neural Information Processing Systems 37 (2024): 24523-24546.

---

### Author Response · Authors · 2025-12-03

Dear Area Chair,

We sincerely thank all reviewers for their constructive feedback. The main concerns raised by all reviewers have now been addressed (e.g., comparison with SigLIP, evaluation of alternative fine-tuning strategies, and generalization verification). Before the rebuttal instructions changed, two reviewers had already taken part in the discussion, and both indicated that the revisions satisfactorily resolved their concerns: **Reviewer W71t raised the score to 6**, and **Reviewer UWJ9 comments "tends to give boarderline accept"**. The other two reviewers have not participated in the discussion, but their concerns have also been addressed. In the following two paragraphs, we outline the paper's main contributions and summarise each reviewer's main concerns along with the actions we have taken.

Our core contribution is a comprehensive and rigorous investigation of CLIP's comprehension of 360-degree textual and visual semantics from a statistical perspective. Our analyses show that while all evaluated CLIP models benefit significantly from explicit textual identifiers, they fail to robustly preserve semantic alignment under horizontal circular shifts, providing the first systematic evidence of this limitation. To address this, we propose a tailored LoRA-based fine-tuning framework that enhances shift invariance while revealing a trade-off between improved robustness and baseline semantic evaluation performance. Through additional rebuttal-phase analyses, we further extend these contributions by studying multiple fine-tuning strategies, evaluating generalization across unseen shift magnitudes and more diverse scenes, and conducting new comparisons with SigLIP to clarify why CLIP remains the de facto evaluator in current 360-degree panorama generation pipelines.

| Reviewer | Core Concern | What We Have Done |
|---------|--------------|-------------------|
| **UWJ9** | Distinctiveness of the 360-degree panorama domain | Clarified why 360-degree panoramas constitute a distinct setting; supported by recent literature. |
| **UWJ9** | Limited methodological novelty | Explained that novelty lies in the fine-tuning objective rather than the adapter; highlighted previously unreported empirical findings and the robustness–performance trade-off. |
| **UWJ9** | Use of CLIP instead of more recent alternatives | Explained the reason to focus on CLIP; added a controlled SigLIP comparison (Sec. J) showing its scores do not reliably reflect image-text semantic alignment . |
| **UWJ9** | Lack of deeper insights | Highlighted new insights: reliance on textual identifiers, instability under circular shifts, and the trade-off between enhanced robustness and original semantic evaluation performance. |
| **W71t** | Different fine-tuning strategies | Added experiments comparing LoRA (image encoder / both encoders) and full fine-tuning (Sec. I). |
| **W71t** | Generalization to unseen shift magnitudes | Added tests across seven unseen shift magnitudes demonstrating robust generalization (Sec. H.2). |
| **W71t** | Effect of pretraining data scale | Compared LAION-400M and LAION-2B pretrained models; conclusions hold consistently (Sec.4 and Sec. F). |
| **3aJi** | Need for more quantitative evidence | Added extensive CLIP-score analyses for textual and visual semantics (Sec. K). |
| **3aJi** | Evaluation on more diverse scene types | Added the new *360_nature* dataset and demonstrated strong cross-scene generalization (Sec. H.1). |
| **3aJi** | Jointly fine-tuning the text encoder | Added experiments on LoRA applied to both encoders and full fine-tuning (Sec. I). |
| **3aJi** | Downstream-task results | Clarified why downstream tasks are beyond the current scope but form an important future direction. |
| **3QfH** | Unclear central focus | Further clarified the core theme of probing CLIP’s comprehension of 360-degree textual and visual semantics. |
| **3QfH** | Whether MLLMs could serve as evaluators | Explained why CLIP (not MLLMs) was investigated; positioned the exploration of MLLMs as future work (Sec. 6).  |
| **3QfH** | Justification of 360-degree data characteristics | Clarified that 360-degree panoramic images are statistically negligible in CLIP’s web-scraped pretraining datasets. |
| **3QfH** | Motivation of stability-bound design | Added intuitive explanation using a simple analogy. |
| **3QfH** | Minor presentation issues (notation, CLIPScore, dataset definitions) | Clarified notation (360×180), added CLIPScore reference, and defined datasets earlier for clarity. |

We hope that these extensive revisions demonstrate the paper's technical rigor, clarity, and contribution to the community. Thank you for your time and careful consideration.

---

### Meta-Review · Area_Chair_6Tzt · 2026-01-03

**Summary:**

After reading the paper and discussions, the AC agrees with the concerns in the review, including 1) the limited novelty of the proposed simple method and intuitive results. 2) the narrow scenario of the task. 3) the lack of deeper insight. Based on the problem, the AC tends to reject this paper

**Reviewer Concerns:**

Most concerns are answered by the authors; the concern about the experiment strategy/metric is well addressed. However, for the more critical novelty and insight, the rebuttal failed to convince the AC

**Reviewer Scores:**

Reviewer W71t raised the score to 6,
Reviewer UWJ9 comments "tends to give a borderline accept."
For reviewer 3aJi, the author failed to provide any demonstration of downstream performance. The score would be unchanged
For reviewer 3QfH, the author also did not provide any experimental results to prove their new claim and discussion. The score would be unchanged

---

### Decision · Program_Chairs · 2026-01-26

Reject